complexity/computer modelling and simulation

agent-based modelling, data assimilation, model calibration, complex systems

**Author for correspondence:**
Le-Minh Kieu
e-mail: m.l.kieu@leeds.ac.uk

# Dealing with uncertainty in agent-based models for short-term predictions

Le-Minh Kieu[1], Nicolas Malleson[1,2] and
Alison Heppenstall[1,2]

[1]University of Leeds, Leeds, UK
[2]Alan Turing Institute, London, UK

L-MK, 0000-0001-7798-6195

Agent-based models (ABMs) are gaining traction as one of the most powerful modelling tools within the social sciences. They are particularly suited to simulating complex systems. Despite many methodological advances within ABM, one of the major drawbacks is their inability to incorporate real-time data to make accurate short-term predictions. This paper presents an approach that allows ABMs to be dynamically optimized. Through a combination of parameter calibration and data assimilation (DA), the accuracy of model-based predictions using ABM in real time is increased. We use the exemplar of a bus route system to explore these methods. The bus route ABMs developed in this research are examples of ABMs that can be dynamically optimized by a combination of parameter calibration and DA. The proposed model and framework is a novel and transferable approach that can be used in any passenger information system, or in an intelligent transport systems to provide forecasts of bus locations and arrival times.

## 1. Introduction

Agent-based modelling (ABM) [1] is a field that excels in its ability to simulate complex systems. Instead of deriving aggregated equations of system dynamics, ABM encapsulates system-wide characteristics from the behaviours and interactions of individual agents, e.g. human, animals or vehicles. ABM has emerged as an important tool for many applications ranging from urban traffic simulation [2], humanitarian assistance [3] to emergency evacuations [4].

Despite the many advances and applications of ABM, the field suffers from a serious drawback: models are currently unable to incorporate up-to-date data to make accurate real-time predictions [5–7]. Models are typically calibrated once, using historical data, then projected forward in time to make a prediction. Here, calibration is ideal for one point in time, but as the simulation

progresses, the prediction rapidly diverges from reality owing to underlying uncertainties [7]. These uncertainties come from *dynamic* (changing over space and time), *stochastic* (containing inherent randomness) and *unobserved* (unseen from the data) conditions of the real system under study. An example of such a system can be found in bus routes. Each time a bus reaches a bus stop, the number of alighting passengers is unknown and the number of waiting passengers downstream is unobserved. The bus route's conditions also change over time, e.g. traffic varies over the route and at different (peak) periods of the day. There are methods that have been developed to incorporate streaming data into models, such as *data assimilation* (DA) routines [8,9]. Broadly, DA refers to a suite of techniques that allow observational data to be incorporated into models [9] to provide an optimal estimate of the evolving state of the system. Performing DA increases the probability of having an accurate representation of the current state of the system, thereby reducing the uncertainty of future predictions. This is a technique that has been widely applied in fields such as meteorology, hydrology and oceanography [10].

There are, however, two methodological challenges that must be overcome to apply DA in ABM. First, DA methods are often intrinsic to their underlying models which are typically systems of partial differential equations with functions linearized mathematically. Hence DA methods typically rely on linearizing the underlying model [11]. One of the most appealing aspects of ABMs is that they are inherently nonlinear, so it is not clear whether the assumptions of traditional DA methods will hold. Second, it is still unknown how much uncertainty DA can effectively deal with when implemented within ABM. Assimilation of real-time data into ABMs has only been attempted a few times and these examples are limited by their simplicity [5–7].

This paper is part of a wider programme of work[1] that is focused on developing DA methods to be readily used in ABM. This paper focuses on one particular model that aims to make predictions of bus locations in real time. Bus route operation has been chosen owing to its inherent uncertainties—for example, a model will need to account for uncertain factors affecting how buses travel on the roads [12]—but also for its tractability—it contains fewer interactions than for example, a model of a crowd. We also focus on one particular DA algorithm—the particle filter (PF). This method is chosen because of its ability to incorporate data into nonlinear models such as ABMs [13].

The objectives of this paper are to: (i) improve the accuracy of short-term forecasts by (ii) performing dynamic state estimation to reduce the uncertainty in the model's estimate of the *current* system state.

All the numerical experiments in this paper will be tightly controlled, following an 'identical twin' experimental framework (for example, see [6]). We will first develop a complex ABM of a bus route to generate fine-grained synthetic global positioning system (GPS) data of buses, that are reasonably similar to real GPS data, for use as synthetic 'ground truth' data. We call this model 'BusSim-truth'. The next step is to develop companion ABMs that are of simpler nature than BusSim-truth. These models do not know the parameters of BusSim-truth nor possess the dynamic and stochastic features of BusSim-truth. We will calibrate and evaluate these companion ABMs against the data generated from BusSim-truth. This experiment is designed to be similar to the real-time monitoring and predictions of bus locations, where models are often a simpler version of reality. The prediction of bus location and arrival times are essential for bus operators and a topical research challenge [14]. The methods developed here can easily be applied to simulation and forecasting for *real* bus systems and could, therefore, offer considerable potential impact. This is particularly pertinent in rapidly developing cities where daily bus schedules can be extremely erratic. In these cases accurate, up-to-date estimates of current waiting times will be highly beneficial to citizens who use (or would like to use) public transport.

The contributions of this paper are threefold. First, several ABMs of bus routes are constructed that account for the interactions between the bus and passengers, the bus and the surrounding traffic, and between multiple buses. While model development is not the sole focus of this paper, these bus route ABMs are novel and have use for other studies. Second, this paper introduces a combination of parameter calibration and DA techniques that can dynamically optimize an ABM to enable accurate estimation of the bus system in real time. Third, this paper shows and quantifies the impacts of calibration and DA in dealing with the stochastic and dynamic nature of the system under study.

This paper is structured as follows. Section 2 describes the research problem and the related works in the literature. Section 3 outlines the methodology. Section 4 describes the numerical experiments that are conducted and discusses these results. Section 5 describes the limitations and implications of the research. Finally, §6 concludes the study and considers the opportunities for future work.

---

[1]See http://dust.leeds.ac.uk/.

# 2. Research problem and related works

Historical and real-time bus GPS data are often used by operators to locate buses and predict their locations and arrival times. The prediction of bus locations and arrival times in real time is a challenging problem [15]. Ideally, perfect knowledge of the current state of the system and any underlying processes is required. However, obtaining this level of knowledge is impossible owing to sources of uncertainty and the complex interactions in bus operations. The majority of research within this area has focused on machine learning methods to find a direct mapping between input data and bus arrival time. Examples of these methods include artificial neural networks [15], support vector machines [14] and Bayesian techniques [12]. While machine learning methods can provide very accurate and efficient predictions in real time, they are solely reliant on the quality and quantity of the available data. Even with high-resolution datasets that record accurate spatio-temporal bus locations, there is an array of additional features that are not recorded in the observed data (such as the downstream population waiting for a bus) so the full complexity of the system will never be captured.

Instead, analytical and simulation models of bus routes have been proposed that aim to reproduce the *underlying processes* in bus operations, rather than attempting to identify direct mappings between inputs and outputs. One of the earliest successes in simulating a simple bus systems was from cellular automata modelling [16–19]. While the dynamical foundations of these models are well understood, they are outperformed by more sophisticated models such as bus-following models [20–24]; traffic-following models [25–27], schedule-following models [28] and ABMs [29,30]. Bus-following models aim to model the fundamental dynamics of a bus route by modelling individual buses that follow each other (for example, speeding up if the bus ahead is far away). Traffic-following models, on the other hand, aim to model buses as a component of a transport system with private and public transport, where their speeds are affected by the traffic flow, traffic signals [26] or traffic density [27]. Schedule-following models assume that buses try to adhere to their schedules and adapt their speeds as they are ahead or behind the schedule [28]. There are also several recent ABMs of public transport, where buses are interacting with other buses and the traffic [29,30]. Compared to other approaches, ABM is able to replicate more complex phenomena in bus' operations, such as crowding, bus bunching and leap-frog bunching [29]. ABMs of buses are also often implemented in advanced ABM systems, such as MatSim [29] and AnyLogic [30], where multiple interactions and heterogeneous behaviour can be modelled.

One way for these models to fit better to the observed data is to adjust the model parameters until the model satisfies some predetermined criteria. This parameter adjustment process is often referred to as *parameter calibration*. Popular optimization techniques include simulated annealing [31], genetic algorithms, [32,33] and approximate Bayesian computation [34]. Parameter calibration, especially with ABMs, is often only implemented once, and therefore cannot account for any changes that may take place within the system during run time. In fact, the real bus operation is certainly *dynamic*, where the system states are changing over time in response to traffic conditions or passenger demand. In real time, there is also uncertainty about the bus operations that comes from the lack of information regarding the current system states. Examples of such unobservable system states include the number of passengers who are waiting at downstream stops or the number who plan to get off the bus, and the surrounding traffic conditions. The lack of information about these factors means that any model of bus operation in real time will have to make assumptions thereby introducing errors in their predictions.

Therefore, this research will develop a framework that allows complex simulation models of buses to deal with uncertainty in real time and make short-term predictions. We will explore a combination of parameter calibration and a DA technique to deal with both the two types of uncertainties about the bus operation that have been discussed: the *changing system states* and the *unobservable information*. The idea is to calibrate an ABM bus route simulator using historical data which is then dynamically optimized on-the-fly using real-time data. This, in itself, is a novel and important contribution. Previous efforts have attempted to incorporate DA with ABMs directly (for example, see [6,7]) and indirectly (e.g. through a 'self-evolving simulation framework' [35]) but it is unclear how DA methods, that have typically been created for linear models [11], can be adapted for nonlinear ABMs.

DA methods assume that observational data are sparse and only describe the target system in limited detail. Therefore, a model is essential as a means of filling in the gaps in space and time left by the observations through the generation of additional data. In effect, the model propagates data from observed to unobserved areas [36]. Although techniques can be used to perform parameter estimation, they are most often framed as a state estimation problem. The aim is to calculate a posterior

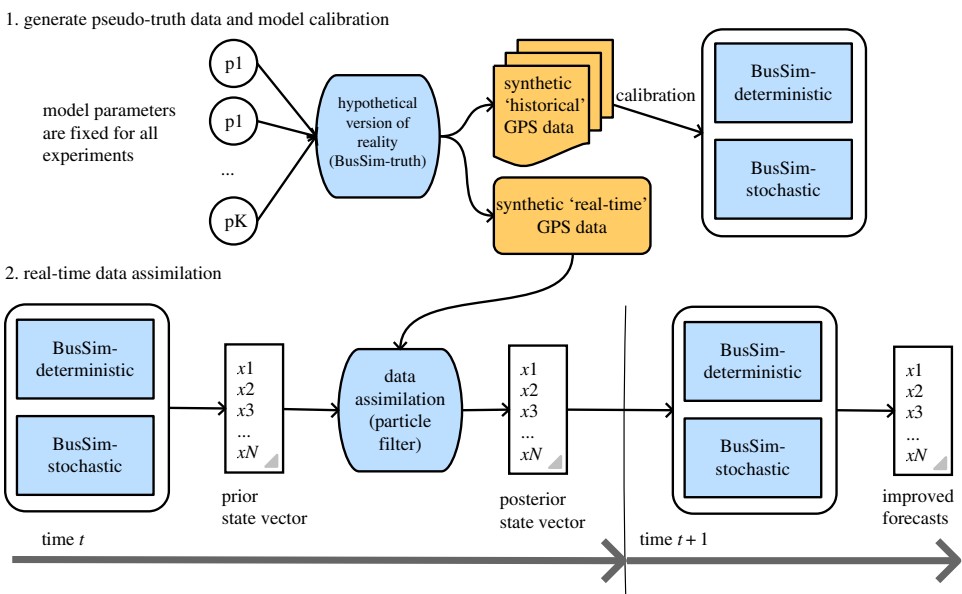

**Figure 1.** Study workflow.

probability for the state vector $X_t$, given prior distributions from a model (in this case, a bus route operation model) and data from observations. It is this marriage of a model and real-time observational data (and the associated uncertainties) that offers the means of allowing all the available information to be used to determine the true state of the system as accurately as possible [37].

Models where the system state at time $t$ are only dependent on the state at time $t-1$ are termed *Markovian*. We are particularly interested in ABMs that can be written in a Markovian nature because DA algorithms require knowledge to the *full* model state in the form of the state vector $X_t$. While some ABMs in the literature track agent histories and use this information to decide future states, these can be recast as Markovian ABMs by expanding the state vector to include these histories. Implementing a bus route system as a Markovian model requires variables such as vehicle locations, speeds, occupancies etc. It is reasonable to assume that the system state at the next time step only depends on the value of these variables at the current time step. For simplicity, we assume that the state vector used here has a fixed size. The unused variables (i.e. those for buses that have yet to enter the system) can be set to zero, enabling the state vector to be treated as sparse and passed efficiently between iterations. If the state vector has a fixed size, then all possible states of the system belongs to a state-space $\mathcal{X} \in \mathbb{R}^n$. The system state evolves in some fixed interval $\{0, \dots, K\}$. We denote the state of the bus route at time $t$ by $X_t \in \mathcal{X}$.

In practice, one would develop simulation models of bus systems that aim to replicate the real bus operations by providing outputs that are as close as possible to some historical bus data. This paper, instead, follows a 'pseudo-truth' experiment framework, similar to [6]. The experiment data to be used will be 'synthetic', or generated from simulation instead of using real data. The reason is that real data often comes with noise that hides the true state of the bus route (e.g. noise from GPS data). A simulated synthetic data would enable us to control the level of noise in the data, and to evaluate the modelling results against the ground truth rather than noisy data. Figure 1 shows the workflow of this study.

Any simulation model, in practice, is essentially an imperfect replication of reality. For instance, the real bus operation is both *dynamic* (the system states are changing over time) and *stochastic* (there is inherent randomness in the system). Recall that the objective of this paper is to improve the accuracy of short-term forecasts using ABMs by performing dynamic state estimation of the current system state. This is essentially the final product at time $t+1$ in figure 1. Improvements of the forecasts at time $t+1$ is archived by improving the estimation of system state at time $t$, by using DA to transform a 'prior state vector' (pure models' estimates) to a 'posterior state vector' (models' estimates combined with real-time data).

Taking this into consideration, we develop a framework (illustrated in figure 1) where the model that will be used to generate the synthetic 'pseudo-truth' data that is both *dynamic* and *stochastic*, to be as close as possible to the real system. We will refer to this model as *BusSim-truth*. The developed framework will

then be evaluated on two simpler variations of this model. This is similar to the practice where no simulation model of bus operations would be able to completely replicate the dynamics of the real bus system. Both simpler models are static (i.e. they both have states that are unchanged over time) but one of them is *deterministic* and another one is *stochastic*:

(i) **BusSim-deterministic**: this model evolves exactly the same way in each model run; and
(ii) **BusSim-stochastic**: this model is stochastic, e.g. the numbers of people waiting at bus stops is drawn from a random distribution.

The study workflow generally consists of two major steps. It starts with the development of the *BusSim-truth* model. Two sets of pseudo-truth data will be generated. The first represents 'historical' GPS data, which are essentially the outputs of multiple runs of the same BusSim-truth model with the same predefined set of parameters. The GPS data will be slightly different each time the model is run because BusSim-truth is stochastic and dynamic. The second set of data represent a single run of BusSim-truth, also using the same set of parameters. These data will represent synthetic 'real-time' GPS data and will be used to conduct DA. This situation is similar to the reality, where transport companies collect data across multiple days to build up a 'history' of the behaviour of the bus system and subsequently use these data to calibrate models. The 'real-time' data represent the *current* state of the world. The BusSim-truth model is not a perfect replication of reality, but contains the *dynamic* and *stochastic* features similar to a real bus system. It also replicates frequent phenomenon in bus operations such as bus bunching (where two buses of the same line arrive at the same bus stop at the same time).

As would be necessary in reality, BusSim-deterministic and BusSim-stochastic will first be calibrated against the synthetic 'historical' GPS data. In the second step of the study workflow, DA will be used in an attempt to update the states of the models to the 'real-time' GPS observations in order to produce more accurate short-term forecasts of the system behaviour.

# 3. Methodology

## 3.1. A hypothetical version of reality: BusSim-truth and its two simpler variations

The first step in the proposed workflow is to develop an agent-based bus route model that will be used to generate synthetic GPS data for each bus on the route (BusSim-truth). BusSim-truth is a stochastic and dynamic model with two classes of agents (bus and bus stop) and predefined parameters (table 1). It is stochastic because the number of boarding passengers is drawn from a random distribution, and dynamic because its parameters gradually change over time. The level of stochasticity and dynamicity in BusSim-truth can also be adjusted to represent bus route systems where conditions are largely stable or volatile over time.

Figure 2 illustrates the workflow for BusSim-truth. Only a brief explanation of the model is included here, as more information on how the BusSim-truth model works can be found in appendix A. At each current time step, each bus agent checks whether the next time step would be larger than the vehicle's scheduled dispatch time. If it is, we then check whether the bus is on the road (status equals MOVING), or at a stop for passenger dwelling (status equals DWELLING), or has finished its service (status equals FINISHED), otherwise the bus remains IDLE.

If the status is MOVING, we first check whether the bus is at a bus stop, by comparing the GeoFence area of each bus stop agent with the bus location. If the bus is not approaching a bus stop, its current speed will be compared with the surrounding traffic speed. In the case it is slower, we assume that the bus will speed up. If the speed already matches the traffic speed, the bus will maintain the same speed. Currently, the traffic volume on the whole network is represented as a single dynamic parameter, although in practice it would be relatively simple to make the traffic volume heterogeneous across the network. The system will first check if the stop is at the last stop when the bus is approaching a bus stop, where the bus status will be changed to FINISHED and the bus speed changed to zero. If it is not the last stop, the system will change the status of agent bus to DWELLING and its speed to zero.

As described in §2, we use BusSim-truth to generate two sets of synthetic data: (i) 'historical' GPS data that simulate normal bus route operation over a number of days and are used for calibration; and (ii) 'real-time' GPS data that represents a single run of the model and are used to represent the

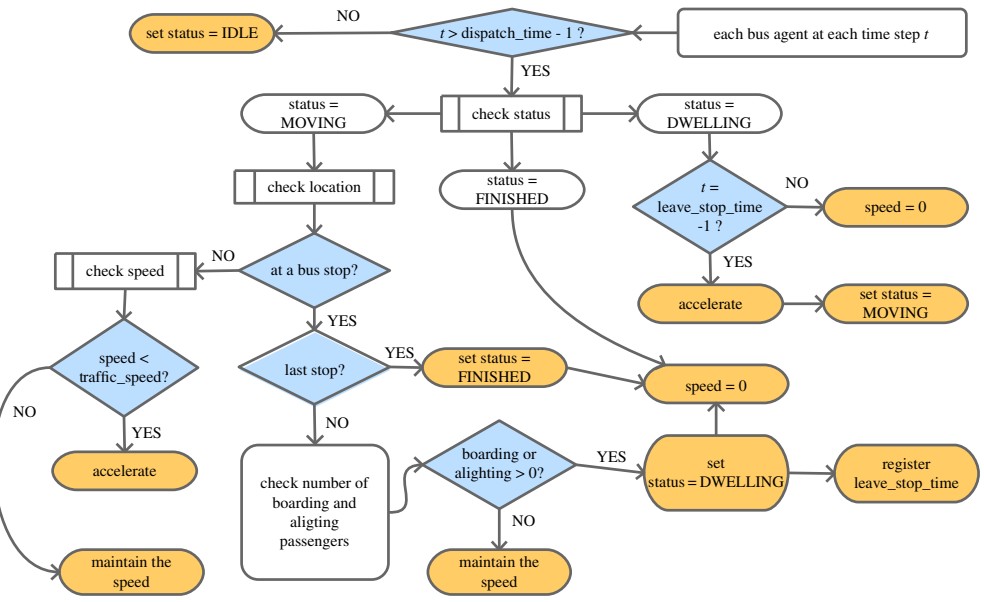

**Figure 2.** Flowchart of BusSim-truth.

**Table 1.** Type of agents and their parameters in BusSim-truth.

| bus agents' parameter/variables | description |
| --- | --- |
| BusID | unique ID of the bus agent |
| acceleration | the acceleration value in m s$^{-2}$ if the bus needs to accelerate |
| StoppingTime | deadtime due to door opening and closing if the bus has to stop |
| visited | list of visited bus stops |
| states $c_j^t$ | whether the bus is idle, moving, dwelling or finished at time $t$ |
| distance travelled $s_j^t$ | coordinate of bus locations on a 1-D lattice at time t |
| $Occ_j^t$ | occupancy of the bus at time $t$ |
| $V^t$ | traffic speed in m s$^{-1}$ at time $t$ |

| stop agents' parameter/variables | description |
| --- | --- |
| StopID | unique ID of the bus stop |
| position | distance from the first stop |
| $Arr_m$ | passengers arrived to the stop per second |
| $Dep_m$ | percentage of onboard passengers alight at the stop |
| arrival_time | store actual arrival time of buses at the stop |
| GeoFence | a circle area to identify whether the bus is at the bus stop |
| $V^t$ | traffic speed in m s$^{-1}$ |

bus system *today*. These are visualized in the time–space diagram in figure 3. Each coloured line shows the trajectory of one bus in the 'historical' GPS data. The bold black lines are another instance of the bus trajectory that we consider as the 'real-time' GPS data.

The *x*-axis shows the time of simulation from 0 to 6000 s, where multiple bus services can be found. The *y*-axis shows the distance from the first bus stop (distance equals zero) to the last stop (distance equals 40 000 m). Assuming that all buses start their service on-time, figure 3 shows bunches of bus service (each with a different colour) and their associated synthetic real-time data (bold lines). As the BusSim-truth model is stochastic, there are a spread of trajectories within each service. This is similar to the reality where buses operate slightly differently on multiple days.

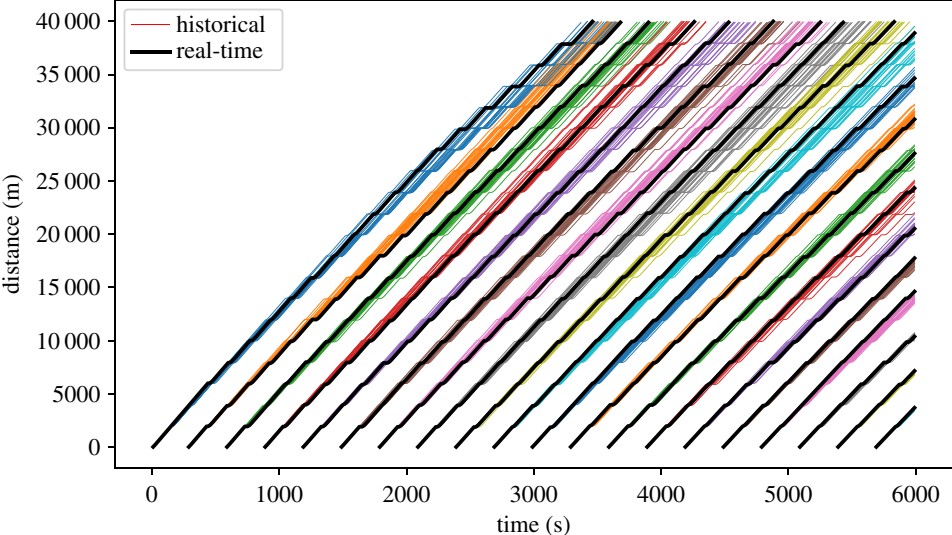

**Figure 3.** Synthetic 'historical' versus 'real-time' GPS bus location data.

Each record in the synthetic data is called an *observation vector*. The vector contains all of the observations made from the 'real world' (in this case the BusSim-truth).

## 3.2. Optimizing the parameters of BusSim-deterministic and BusSim-stochastic

Most ABMs have a large number of parameters. For the BusSim models, the model parameter vector $S_t$ at time $t$ contains the arrival rate $\mathrm{Arr}_m^t$, departure rate $\mathrm{Dep}_m^t$ at each stop $m$, and the traffic speed $V^t$:

$$S_t = \begin{bmatrix} \mathrm{Arr}_m^t & \mathrm{Dep}_m^t & V^t \end{bmatrix} \quad m = 1 \ldots M. \tag{3.1}$$

The two simpler models (BusSim-deterministic and BusSim-stochastic) first need to be calibrated to the observations (the historical data). Here, an automatic parameter calibration process, based on the cross-entropy method (CEM) [38] is used. CEM is a population-based Monte Carlo learning algorithm to combinatorial multi-extremal optimization and importance sampling. It originated from the field of rare event simulation, where even small probabilities need to be estimated [38]. In principle, CEM develops a *probability distribution* over possible solutions for the optimal parameters of the model. New solution candidates are drawn from this distribution and are evaluated. The best candidates are then selected to form a new improved probability distribution of the optimal parameters, until certain criteria are met. CEM is chosen over other popular optimization methods in parameter calibration of ABMs, such as genetic algorithm [32] and simulated annealing [31], because of its probabilistic nature that facilitates the calibration of stochastic models [39]. The interested reader may refer to [38], and various applications of CEM, such as [39], for a more detailed account. A pseudo-code of the CEM algorithm that we adopted for this paper has also been described in appendix B.

Formally, the parameter calibration is an optimization problem to minimize some performance index $\mathrm{PI}(\pi)$ over all $\pi \in \mathbb{R}^k$. Here, a solution $\pi = (\pi_1, \pi_2, \ldots, \pi_k)$ denotes a set of parameters of the model under consideration and $k$ denotes the number of dimension in this set (see equation (3.1)). Let $\pi_*$ denote the optimal solution, or the best set of model parameters that we want to find, that is

$$\pi_* = \mathrm{argmin} \quad \mathrm{PI}(\pi), \quad \pi \in \mathbb{R}^n. \tag{3.2}$$

The above objective function is equivalent to finding $\pi_*$ such that $\mathrm{PI}(\pi_*) \leq \mathrm{PI}(\pi) \; \forall X \in \Pi$, where $\Pi$ is a constrained parameter space such that $\Pi \in \mathbb{R}^k$. The performance index $\mathrm{PI}(\pi)$ is generally the difference between model output and observed data. The complexity of this problem comes from the stochasticity of BusSim, where the same solution $\pi$ may yield a different realization $\mathrm{PI}(\pi)$. To reduce this stochastic effect, it is necessary to run the (stochastic) model multiple times, and to evaluate the simulation outputs against a compilation of observed data from multiple days or instances. Let $K_I$ be the number of replications required for each model evaluation and $K_O$ be the

number of instances in the observed data, we can derive a more detailed objective function of the parameter calibration problem:

$$\min \mathrm{PI}(\pi) = \frac{1}{N \cdot T} \sum_{t=1}^{T} \sum_{n=1}^{N} \left[ \left| \frac{1}{K_I} \sum^{K_I} s_{j,i,t}^{\mathrm{SIM}} - \frac{1}{K_O} \sum^{K_O} s_{j,o,t}^{\mathrm{OBS}} \right| \right.$$
$$\left. + \left| \sqrt{\frac{\sum^{K_I} \left( s_{j,i,t}^{\mathrm{SIM}} - \hat{s}_{j,i,t}^{\mathrm{SIM}} \right)^2}{K_I - 1}} - \sqrt{\frac{\sum^{K_O} \left( s_{j,o,t}^{\mathrm{OBS}} - \hat{s}_{j,o,t}^{\mathrm{OBS}} \right)^2}{K_O - 1}} \right| \right], \tag{3.3}$$

where $N$ is the number of buses, $T$ is the number of time steps, $s_{j,i,t}^{\mathrm{SIM}}$ is the location of simulated bus agent $j$ at time $t$ for the replication $i$, and similarly $s_{j,o,t}^{\mathrm{OBS}}$ is the synthetic observed location of bus $j$ at time $t$ for the instance $o$. The objective function in equation (3.3) can be seen as the sum of the difference in mean location and standard deviation of locations at each time step for each bus and each replication/instance between the simulated outputs and synthetic observed data. Owing to the stochasticity within the system, we need to evaluate the difference in the mean and standard deviation of bus locations (the spread of bus locations over multiple instances are important).

## 3.3. Data assimilation using a particle filter

We can formulate an ABM as a state-space model $\dot{X}_t = f(X_t) + \epsilon_t$ and use DA to dynamically optimize the model states and parameters with up-to-date data to reduce uncertainty. The state-space model is represented by a state-space vector $X_t$ at time $t$, which contains all information of the current state of each agent in the model, and the important parameters that we want to dynamically optimize as the model is running:

$$X_t = [O_t \quad S_t]$$
$$= \begin{bmatrix} c_j^t & s_j^t & v_j^t & \mathrm{Occ}_j^t & \mathrm{Arr}_m^t & \mathrm{Dep}_m^t & V^t \end{bmatrix}. \tag{3.4}$$

The state-space vector $X_t$ must contain all of the information that identifies the current state of the modelled system, allowing it to be projected forward to the next time step. Thus vector $X_t$ usually contains both the observation vector $O_t$ and the model parameters vector $S_t$ (equation 3.1) at time $t$. Note that $S_t$ has been calibrated in the previous section, but is still included in the state-space vector $X_t$ to allow the model to be dynamically optimized with new data—this is essential in dynamic situations where parameter values change over time. This approach is often referred to as dynamic calibration [40].

DA is a suite of methods to adjust the state of a running model using new data to better represent the current state of the system under study [7]. DA was born out of data scarcity, where observation data are sparse and insufficient to describe the system. Notwithstanding the proliferation of new data sources, insufficient data is still a major problem in research. The prediction of bus locations is a clear example where the number of future boarding and alighting passengers are unknown in real time. DA algorithms fill in the spatio-temporal gaps in the observed data by running a model forward in time until new observed data are available. This is typically called the *predict* step in DA algorithms. After the predict step, DA has an estimate of the current system state and its uncertainty (which is often referred as the 'prior' in Bayesian literature). The next step is typically called the *update* step, where new observations and uncertainty are used to update the current state estimates. The result is referred to as the 'posterior' in Bayesian literature, and should be the best guess of the system state from both the observations and model.

There are several DA algorithms in the literature, ranging from the simple Kalman filter [41] to more advanced extensions, including extended, ensemble and unscented Kalman filter [7]. These algorithms generally aim to extend the original Kalman filter by relaxing the assumption of linearity and introducing methods to work with nonlinear models. However, they may not be the most suitable candidate to incorporate data into ABMs for two reasons. First, ABMs are driven by a large number of interacting agents with goals, history and behavioural rules. As a result, they lack an analytic structure, such as differential or difference equations, to facilitate the implementation of the Kalman filter and its extensions where often the model Jacobian and covariance matrices need to be formulated [6]. Second, although the assumption of linearity has been relaxed, these extensions assume that the noise in the model estimation is Gaussian.

There is a flexible Bayesian filtering method that has been designed to work with nonlinear, non-Gaussian models without analytical structure; this is the PF. The key idea is to approximate a

**Table 2.** Fixed parameters in BusSim-truth.

| class | parameter | value |
| --- | --- | --- |
| Bus | FleetSize | unique ID of the bus agent |
| | acceleration | 3 m s$^{-2}$ |
| | $[\theta_1, \theta_2, \theta_3]$ | [3, 1, 0.85] s |
| BusStop | number of stops | 20 |
| | length between stops | 2000 m |
| | GeoFence | 50 m |

posterior distribution by a set of samples or particles, drawn from this distribution. Each particle is a concrete hypothesis of the true system state. The set of particles approximates the posterior distribution. PF is best described as a non-parametric Bayes filter because it develops the belief using a finite number of samples.

Hypotheses of the system state at time $t$ is represented by a set $P_t$ of $N_P$ weighted random particles:

$$P_t = \{\langle X_t^{\lfloor i \rfloor}, w_t^{\lfloor i \rfloor} \rangle \mid i = 1, \ldots, N_P\}, \tag{3.5}$$

where $X_t^{\lfloor i \rfloor}$ is the state vector of the $i$-th particle and $w_t^{\lfloor i \rfloor}$ is the corresponding weight. Weights are non-zero, and the sum over all weights is 1. The core idea of the PF is to update and maintain this set of particles given model outputs and observations. A PF recursively estimates the particle set $P_t$ based on the estimate $P_{t-1}$ at the previous time step, and the observation. The PF algorithm can be briefly described in three steps:

  (i) **predict:** generate the next set of particles $\hat{P}_t$ from the previous set $P_{t-1}$. This represents the *prior* distribution to describe how the system state evolves;
 (ii) **importance weighting:** compute the importance weight $w_t^{\lfloor i \rfloor}$ for each particle in $P_t$. This is equivalent to the 'Update' step in the Kalman filter, and will give us the *posterior* distribution; and
(iii) **resampling:** this step has no analogous step in the Kalman filter and its extensions. The resampling step creates a new set of particles from the current set. The likelihood to draw a particle is proportional to its weight. We adopt sample importance resampling (SIR), a popular bootstrap systematic resampling in the PF literature [6,36]. SIR has been developed to deal with *particle deprivation*, which is the problem when particles converge to a single particle after several iterations owing to one particle outperforming all others [42]. This problem significantly reduces the area of state space covered by the particles in later iterations.

Because resampling will generate particles using the existing pool of particles, it will not be able to produce particles where the prediction accuracy is better than the existing particle pool. This means that in *classical* PF, the model parameter $S_t$ of both BusSim-deterministic and BusSim-stochastic will be unchanged over time. Because the parameters change over time, we need to dynamically optimize $S_t$. This problem is solved in this paper by a simple and generic solution. We improve the quality of the particles by diversification similar to [43], in a process also known as roughening, jittering and diffusing [44]. This is achieved by adding a random Gaussian white noise $\sigma$ with mean 0 and a predefined standard deviation, not to the whole state vector $X_t$, but to the model parameter $S_t$, to increase the probability of having particles that represent the current state of the underlying model. The PF is applied to BusSim-deterministic and BusSim-stochastic using up-to-date data from the synthetic 'real-time' GPS data. DA could take place after any number of prior model iterations (e.g. [45] use a DA window of 100 iterations) but here it takes place after every model iteration. This is a fair assumption for this application because in reality bus GPS data could be available in near real-time, or at least at a frequency that is similar to the real amount of time represented by a single model iteration.

# 4. Numerical experiment

## 4.1. Experiment set-up

To generate the synthetic 'historical' and 'real-time' GPS data used in BusSim-deterministic and BusSim-stochastic, we predetermine a set of model parameters to generate realistic GPS data. Table 2 lists the fixed parameters being used in this experiment.

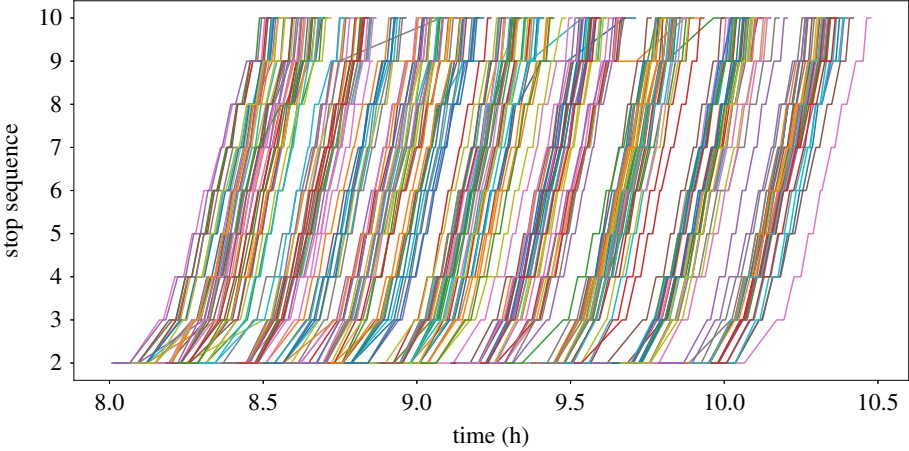

**Figure 4.** Observed bus trajectories of Route 555 in Brisbane, Australia.

Second, the dynamic parameter set $S_t = [\text{Arr}_m^t \ \text{Dep}_m^t \ V^t]$ (equation (3.1)) are time-varying and therefore, randomly generated using fixed rules. We first generate an initial arrival rate $\text{Arr}_m^0$ at stop $m$ at time 0 by a random generation from an uniform distribution between the minimum and maximum passenger arrival rate [minDemand, maxDemand]:

$$\text{Arr}_m = \mathcal{U}(\text{minDemand}, \text{maxDemand}) \quad m = 1, \dots, M. \tag{4.1}$$

The departure rate is also generated from an uniform distribution, but also ordered ascending to represent the fact that more passengers alight at the end of the route than at the beginning. The departure rate at the last stop (stop M) is set as 1 to allow remaining passengers to alight the bus at the last stop:

$$\text{Dep}_m = \text{ordered} (\mathcal{U}(0.05, 0.5)) \quad \text{and} \quad \text{Dep}_M = 1 \ \& \ m = 1, \dots, M. \tag{4.2}$$

Finally, the initial traffic speed is set at $14 \text{ m s}^{-1}$.

## 4.2. The stochastic and dynamic nature of BusSim-truth

This section aims to provide a simple verification that demonstrates BusSim-truth generates realistic synthetic GPS data under different sets of parameters. First, let us look at some real trajectories from observed GPS data, as illustrated in figure 4. The figure shows six months of real GPS trajectories from July to December 2015. The data are from Route 555 in Brisbane, Australia, a busy bus route connecting Logan City to Brisbane Central Business District via a segregated busway. Each GPS record includes information about each visit to a bus stop, including: route number, trip ID, vehicle ID, scheduled departure time, observed arrival time and observed departure time.

Figure 4 shows similar trajectories to the simulated trajectories illustrated earlier in figure 3. As previously discussed, we will not attempt to calibrate and validate BusSim-truth against this observed data. This section rather aims to show that BusSim-truth is reasonably realistic, can replicate common phenomena in bus operations and has parameters that we can easily control for in subsequent experiments.

We aim to control the stochastic and dynamic level in BusSim-truth using only a single parameter for each. Equation (4.1) controls the level of stochasticity in BusSim-truth. For instance, a pair of values [minDemand, maxDemand] = [0.5, 1] means 0.5 to 1 passenger arriving at the bus stop each minute. By fixing the minDemand to be a small number (e.g. equals 0.5), we can control the stochasticity of BusSim-truth by a single parameter maxDemand, with a larger maxDemand meaning more stochasticity and vice versa. We control the level of dynamicity in BusSim-truth by a dynamic change rate parameter $\xi$, which gradually changes the arrival rate and surrounding traffic speed over the simulation period.

To implement an inner verification of the BusSim-truth model and to investigate the impacts of the *stochastic* and *dynamic* natures of the system under study, we evaluate the outputs from BusSim-truth using different values of stochasticity and dynamicity. Figure 5 gives an insight into the differences in bus trajectories when maxDemand equals 0.5 and 2. Note that when maxDemand equals 0.5, BusSim-truth reduces to a deterministic model (similar to BusSim-deterministic). This is because maxDemand would be equal to minDemand.

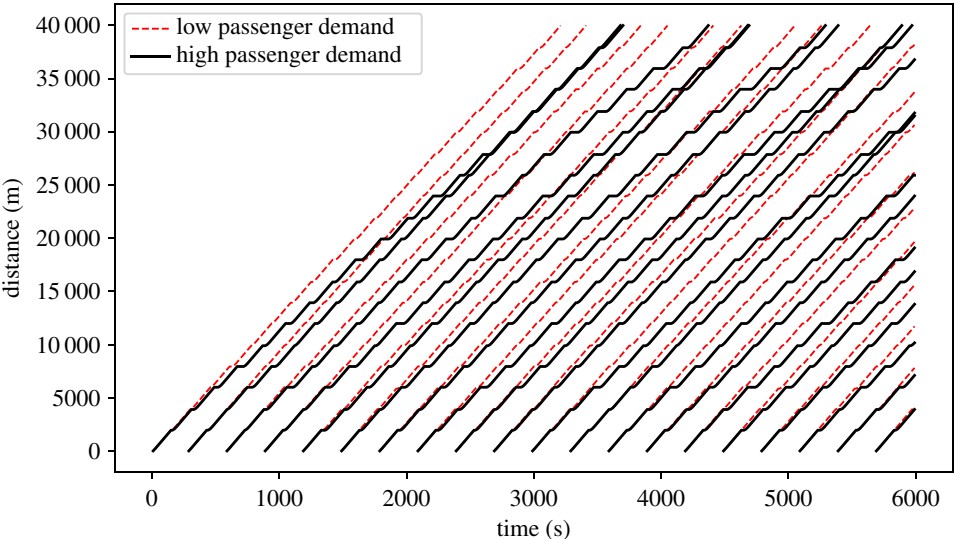

**Figure 5.** Synthetic bus GPS trajectory at low and high passenger demand. Red, dashed lines are bus trajectories when maxDemand equals 0.5, while black, solid lines are bus trajectories when maxDemand equals 2.

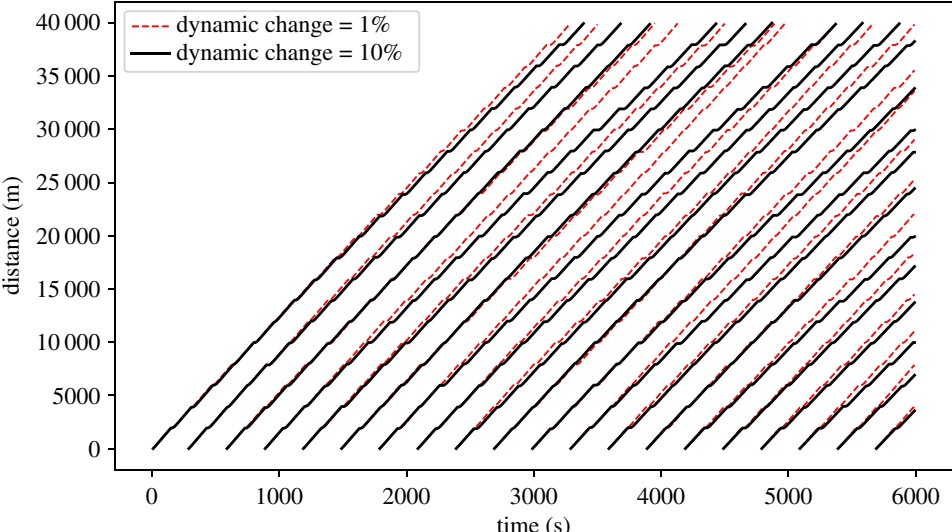

**Figure 6.** Synthetic bus GPS trajectory with two different values of $\xi$.

Each line in figure 5 shows the GPS trajectory of the bus location, as generated by BusSim-truth. The solid lines show the trajectory of buses at high and stochastic demand (maxDemand equals 2), whereas the dashed lines are for low and deterministic demand (maxDemand equals minDemand and equals 0.5). The trajectories in figure 5 show that as the maxDemand increases, there are more delays for individual buses and it is less likely that buses are able to keep stable headway from each other. In times of high passenger demand, BusSim-truth can replicate common phenomena in bus operations, including bus bunching and leap-frog bus bunching.

The dynamic nature of BusSim-truth is illustrated in figure 6 when the dynamic change rate parameter $\xi$ is equal to 1% and 10%. Because the arrival rate and traffic speed gradually change, there is little change in the bus trajectories of BusSim-truth with $\xi$ equalling 1% and 10%. As time passes, there are more delays for BusSim-truth as $\xi$ equals 10%. This is because there are more passengers (higher arrival rate) and the buses are travelling slower (lower traffic speed).

## 4.3. Scenario 1: no calibration (benchmark)

This scenario aims to evaluate the prediction results from BusSim-deterministic and BusSim-stochastic *without* calibrating their parameters, performing DA. This is done to allow evaluation of the predictive

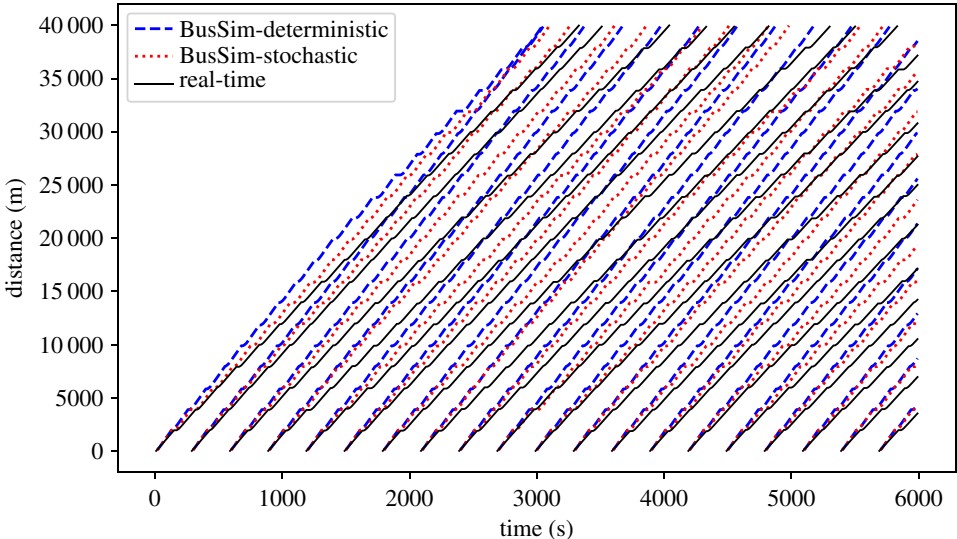

**Figure 7.** Prediction results from scenario 1: no calibration.

performance of the model after calibration *and* DA. The two models are implemented using random parameters generated from equations (4.1) and 4.2. The outputs from these models are bus locations at each time step $t$, which can be compiled to space–time trajectories and compared to the synthetic 'real-time' bus trajectories, as illustrated in figure 7.

Figure 7 shows one particular case where maxDemand equals 2, and $\xi$ equals 7%, as an example of the prediction results. The dashed lines show the predicted trajectories of each bus (each line represents a single bus) from BusSim-deterministic and BusSim-stochastic, whereas the solid line shows the synthetic real-time data. The models are used to make predictions of bus locations at the next time step (next 10 s) without any model training and calibration.

Both models poorly predict the trajectories of the 'real' buses. The gaps between the predictions and the real trajectories widen as the buses operate (as the distance and time increase). This shows that the models are diverging from reality. This is expected because the models do not have the optimal parameters to capture the bus route operations. These models are therefore not useful for real-time prediction without parameter calibration or DA.

## 4.4. Scenario 2: parameter calibration

In this scenario, BusSim-deterministic and BusSim-stochastic are calibrated using the cross-entropy method, as described in §3.2. The two calibrated models are used to predict the bus locations at each time step $t$, which can be compiled to trajectories. Figure 8 shows an example of the comparison between the predictions from BusSim-deterministic and BusSim-stochastic versus the synthetic 'real-time' GPS data, where the maxDemand equals 2 and $\xi$ equals 7%.

Figure 8 shows that both models outperform the models in scenario 1 (no calibration (figure 7). At the first quarter of the route (when Distance is less than 1000), there are few visible gaps between the predicted trajectories and the synthetic real-time data. However, the increasing divergence is still a major problem in this scenario. The gaps between the predicted trajectories and synthetic real-time trajectories widen by time. This is because BusSim-stochastic and BusSim-deterministic are both also *static* models, i.e. their model states do not change over time, whereas the synthetic data comes from a dynamic system. At $\xi$ equals 7%, both the passenger demand and the traffic speed are changing rapidly. The two models have been well calibrated to the synthetic 'historical' data, but will provide a prediction result that is unchanged over time, and fail to take into consideration the dynamical changes in the system states.

Recall that there are differences between the 'historical' and 'real-time' data owing to the stochastic nature of the system under study (figure 3). However, BusSim-stochastic, despite being able to produce stochastic outputs, shows no improvement in prediction performance in comparison to BusSim-determinstic (this is only capable of producing deterministic outputs). This shows that the changing system state is the main source of prediction error, not the deterministic or stochastic nature of the models. To remedy this, we need a procedure that would help the *static* models to deal with

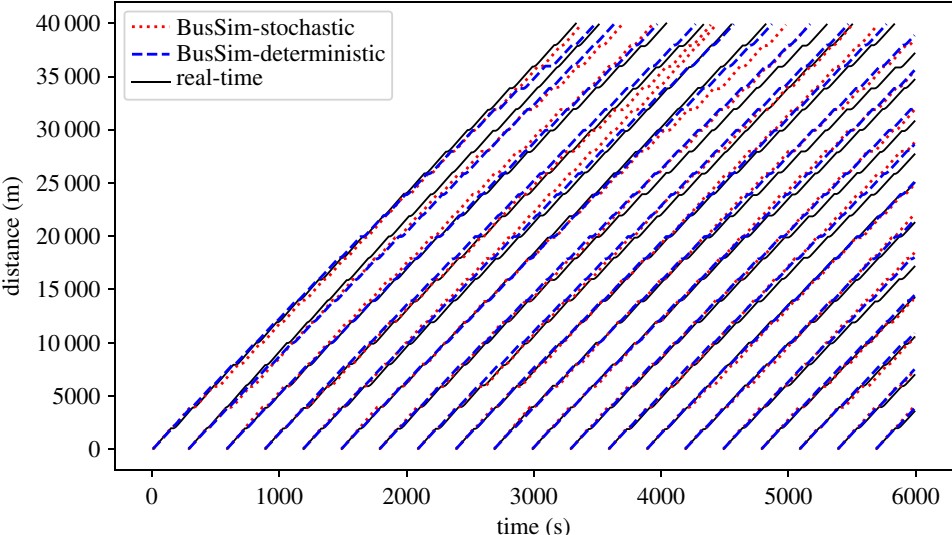

**Figure 8.** Prediction results from scenario 2: parameter calibration.

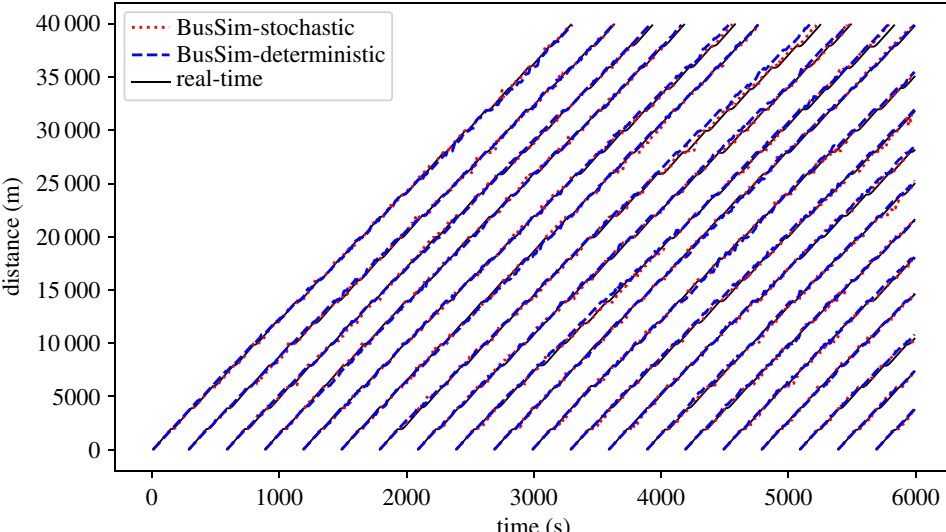

**Figure 9.** Prediction results from scenario 3: parameter calibration and particle filtering.

the uncertainty from the changes in a dynamic system and prevent the errors gradually increasing throughout the simulation.

## 4.5. Scenario 3: applying a particle filter

As discussed, DA is the chosen approach to enable the static BusSim-stochastic and BusSim-deterministic models to deal with the uncertainty from a system that is changing over time. PF is the specific DA algorithm being adopted in this paper. PF is able to deal with nonlinear, non-Gaussian models without an analytical structure. The well-known high computation cost concern of the PF [6,33] is not an issue in this study because of the limited number of agents used (bus and bus stop agents).

This section applies a PF to the calibrated BusSim-deterministic and BusSim-stochastic, as described in §3.3. At each time step $t$, the two models are only provided with the observation vector $O_t$, which uses $O_t$ to correct their prediction of future state vectors $X_t$ to $X_T$, where $T$ is the last time step. Figure 9 illustrates the results after the models have been calibrated *and* have 'real-time' data incorporated (assimilated) into them during runtime.

Figure 9 shows a much better prediction performance, in comparison to the previous scenarios (figures 7 and 8). There are still observable gaps between the prediction and the synthetic 'real-time' GPS data. This is because the underlying models do not know the underlying stochasticity and

**Table 3.** Sensitivity analysis of maxDemand and dynamic change rate $\xi$.

| | values | scenario 1 | scenario 2 | scenario 3 |
|---|---|---|---|---|
| maxDemand | 0.5 | 302 | 102 | 24 |
| | 1 | 313 | 107 | 25 |
| | 1.5 | 319 | 112 | 35 |
| | 2 | 335 | 125 | 49 |
| | 2.5 | 340 | 119 | 52 |
| | 3 | 337 | 127 | 62 |
| | 3.5 | 346 | 133 | 66 |
| | 4 | 338 | 148 | 59 |
| | 4.5 | 341 | 145 | 55 |
| dynamic change rate | 0 | 197 | 75 | 41 |
| | 2.5 | 203 | 77 | 44 |
| | 5 | 208 | 82 | 40 |
| | 7.5 | 211 | 89 | 39 |
| | 10 | 218 | 90 | 49 |
| | 12.5 | 220 | 93 | 47 |
| | 15 | 232 | 97 | 45 |
| | 17.5 | 235 | 102 | 49 |

dynamicity in the synthetic data, but the improvements (which will be quantified shortly) certainly *appear* to be substantial.

Both the models are now capable of providing short-term predictions that are very close to the synthetic real-time data. Similar to the previous scenario, it is also noted that the two models show a similar predicting performance, albeit one is deterministic and the other is stochastic. The PF algorithm has effectively optimized the models as they operate and enabled them to provide good predictions of the bus locations in real time.

## 4.6. Sensitivity analysis

In this section, we perform a sensitivity analysis to compare the prediction error in each scenario. The same experiments, as described in scenarios 1–3, are repeated at different values of maxDemand and dynamic change rate $\xi$. To increase the robustness of the comparison, 10 replications have been made for each experiment, and the average root mean squared error (RMSE) values are reported. RMSE is calculated as the difference in prediction bus location and synthetic 'real-time' bus location:

$$\text{RMSE} = \sqrt{\frac{1}{T}\sum_{k=1}^{T}\left(\hat{y}_k - y_k\right)^2}, \qquad (4.3)$$

where $\hat{y}_k$ and $y_k$ are the bus location at time $k$ from the model prediction and synthetic 'real-time' data, respectively. Table 3 compares the RMSE from each scenario. It is clear that scenario 3 (combination of parameter calibration and DA) outperforms the other two scenarios.

## 5. Implications and limitations

This paper presents an integrated framework to reduce uncertainty in ABMs when making predictions in real time, by combining parameter calibration and DA. As discussed in §§1 and 2, an 'identical twin' approach has been adopted instead of real noisy data to facilitate an effective evaluation of the proposed methods against the synthetic 'ground truth'. The numerical experiment shows that the framework yields more accurate predictions than (i) a benchmark scenario (without parameter calibration), and (ii) a scenario with parameter calibration but without DA.

In its current form, the framework can provide *real time* bus locations and arrival times for passenger information systems. The forecasted bus location and arrival information provides key intelligence for waiting passengers [46]. This is beneficial for all public transport passengers, but can be of particular benefit in countries, for example, in the Global South [47], where there are frequent delays owing to transport systems being complex, heterogeneous or heavily congested. The prediction of bus arrival times is also critical for real-time trip planners. These planning systems propose optimal alternative routes for passengers, or update information on a connecting service that may be unreachable owing to delayed buses.

Many advanced intelligent transport system applications rely heavily on predictions of bus location and arrival times, for example bus control studies such as [48]. A model-based prediction of bus location and arrival time, such as the framework in this paper, would allow bus operators the ability to evaluate and update their transportation infrastructures in real time. Although the proposed *BusSim-Truth* simulation model is relatively simple (at least when compared with industry-leading models such as MatSim [2]) it is able to recreate some of the important features of the bus system such as leapfrogging, bus bunching, and responses to dynamic passenger demand. It is, therefore, adequate as a test bed for the PF; future work will test the PF on a more advanced and realistic simulation.

A notable limitation with the current framework is with regards to the treatment of traffic. Currently, the traffic volume on the network is represented with a single parameter ($V^t$ in equation (3.1)). Therefore, although the traffic volume can vary dynamically, it is homogeneous across the bus route. Although it is technically simple to implement homogeneous (per link) traffic volumes, this would substantially increase the size and complexity of the model state space and would require a much larger number of particles to adequately capture the additional complexity. A further exciting opportunity for future development is to explore the implications of the buses affecting the surrounding traffic, rather than the impact of the traffic on the buses.

DA methods, similar to many other data-driven methods, require and rely on the quality of data. Another limitation of this work is with regard to the cases where the data have major discrepancy, that have not been addressed in this current framework. The current models, in principal, should still produce particles that are more reliable than data that are highly inaccurate. Future studies may look at developing a data quality check procedure into the framework in figure 1 to allow skipping the DA step if the data quality deteriorates to a certain value.

# 6. Conclusion

This paper proposes parameter calibration and DA frameworks to enhance the prediction accuracy in ABMs when the system under study has a *stochastic* and *dynamic* nature. Ultimately the methods will be applied to real data from real systems, but currently hypothetical 'pseudo-truth' data are generated to test the algorithms in an experimental environment as per the 'identical twin' approach. We first develop a stochastic and dynamic ABM of a bus route, referred to as *BusSim-truth*. This model is employed to generate synthetic 'historical' and 'real-time' GPS data of bus locations. The 'historical' data are used to train two simpler models of bus routes, referred to as *BusSim-deterministic* and *BusSim-stochastic*, and evaluated against the 'real-time' data.

Similar to real practice, when any simulation model is a simplification of the reality, BusSim-deterministic and BusSim-stochastic are simpler than BusSim-truth, and thus may not be able to produce a prediction similar to the synthetic 'real-time' GPS data under limited data. We propose a solution for this issue by parameter calibration using cross-entropy method (scenario 2), by a combination of parameter calibration and PF (scenario 3), and show that they outperform the no calibration scenario (scenario 1), at various levels of uncertainty.

This paper shows the need for parameter calibration and DA, and particularly the combination of them, to improve the accuracy of model-based prediction using ABMs in real time. Future research directions includes fitting the proposed framework with real data instead of synthetic data.

Data accessibility. Data and relevant code for this research work are stored in GitHub: https://github.com/leminhkieu/Bus-Simulation-model and have been archived within the Zenodo repository: https://doi.org/10.5281/zenodo.3549633.
Authors' contributions. L.-M.K. developed the bus simulation model and drafted the manuscript. N.M. planned the overall objectives of the paper and the integration of this work to the main research project on data assimilation of ABMs. A.H. wrote the introduction, implications and limitations, and majorly contributed to writing other sections in the manuscript.

Competing interests. We declare we have no competing interests.

Funding. This project has received funding from the European Research Council (ERC) under the European Union Horizon 2020research and innovation programme (grant agreement no. 757455), a UK Economic and Social Research Council (ESRC) Future Research Leaders grant no. (ES/L009900/1) and an ESRC/Alan Turing Joint Fellowship (ES/R007918/1).

Acknowledgements. The authors express their gratitude to Dr Kevin Minors and Mr Andrew A West who significantly contributed to the simulation source code of this work.

# Appendix A. The BusSim model

The following notation will be used to formalize each of the simulation models:

— d$t$: time interval ($s$)
— $t$: current time ($s$ from 00:00:00)
— $N$: number of buses
— $j$: index of vehicle ($j = 1 \ldots N$)
— $m$: index of bus stop ($m = 1 \ldots M$)
— $M$: number of bus stops
— $c_j$: current status of bus $j$ (IDLE, MOVING, DWELLING or FINISHED), where IDLE refers to out-of-service buses and DWELLING refers to buses that are serving passengers at stops.
— $a_j$: acceleration rate of bus $j$ (m s$^{-2}$)
— $v_j$: current speed of bus $j$ (m s$^{-1}$)
— Occ$_j$: current occupancy of bus $j$ (number of passengers onboard)
— $H$: scheduled headway ($s$)
— $C$: bus capacity
— $V$: traffic speed (m s$^{-1}$)
— $\delta_j$: the scheduled dispatch time of bus $j$ from the first stop
— $t_{j,m}^a$: arrival time of bus $j$ at stop $m$
— $t_{j,m}^d$: the moment when bus $j$ can leave stop $m$ after passenger boarding and alighting (or Leave_stop_time)
— $D_{j,m}$: dell time of bus $j$ at stop $m$ for passenger boarding and alighting
— $\theta_1$, $\theta_2$, $\theta_3$: parameters set for estimating $D_{j,m}$
— $B_{j,m}$: number of boarding passenger to bus $j$ at stop $m$
— $A_{j,m}$: number of alighting passenger from bus $j$ at stop $m$
— Arr$_m$: arrival rate of passengers to stop $m$ per second
— Dep$_m$: departure rate of passengers from stop $m$

Figure 2 illustrates the workflow for BusSim-truth. At each current time step $t$, each Bus agent checks whether the next time step would be larger than the vehicle's scheduled dispatch time $\delta_j$. If $t > \delta_j$, we then check whether the bus is on the road (status equals MOVING), or at a stop for passenger dwelling (status equals DWELLING), or has finished its service (status equals FINISHED), otherwise the bus remains IDLE.

If the status is MOVING, we first check whether the bus is at a bus stop, by comparing the GeoFence area of each bus stop agent with the bus' location. If the bus is not approaching a bus stop, its current speed $v_j$ will be compared with the surrounding traffic speed $V$. If $v_j < V$, we assume that the bus will speed up with an acceleration rate $a_j$, thus we have

$$v_j^t = v_j^{t-dt} + a_j \cdot dt. \tag{A 1}$$

Therefore, for the next time step, the bus will cover a distance of

$$S_j^t = S_j^{t-dt} + v_j^t \cdot dt. \tag{A 2}$$

If the speed already matches the traffic speed $V$, the bus will maintain the same speed. Or else if the bus is approaching a bus stop, the system will first check if the stop is the last stop. If it is the last stop, then the bus' status will be changed to FINISHED and bus speed is changed to zero. If it is not the last stop, the system will change the status of agent bus $j$ to DWELLING and its speed to zero. The number of boarding and alighting passengers from the bus $j$, and the time that it will leave the stop are estimated as follows.

The number of boarding passenger is proportional to the time gap between the current time (when bus $j$ approaches the bus stop $m$) and the last time any bus visits the bus stop $m$:

$$B_{j,m} = \lfloor \text{Po}(\text{Arr}_m \cdot (t^a_{j+1,m} - t^a_{j,m})) \rceil \quad | \quad B_{j,m} \in \mathbb{N}. \tag{A 3}$$

Equation (A 3) shows that the number of boarding passengers is estimated using a stochastic Poisson process. A Poisson process is widely adopted in the literature to estimate the count of passengers waiting at a public transport stop [25,27]. Extensions of this stochastic process have been introduced, such as a non-homogeneous Poisson process [49], where the arrival rate is time-dependent, but for simplicity we adopt a homogeneous Poisson process for this paper. Equation (A 3) makes the BusSim-truth model stochastic, because there is randomness in the way the Poisson process generates a number. For more details on the number generation process using stochastic Poisson process (e.g. thinning algorithm), interested readers may refer to [50]. The number of boarding passengers is also limited by the available capacity of the bus:

$$B_{j,m} = \max(B_{j,m}, C - Occ_m). \tag{A 4}$$

The number of alighting passengers is proportional to the number of passenger on board (bus occupancy) and the departure rate at the stop $m$. For simplicity, we assume that $A_{j,m}$ is the product between the departure rate from bus stop $m$ and the current bus occupancy (the number of passenger on board leaving the last stop):

$$A_{j,m} = \lfloor \text{Dep}_m \cdot Occ_{j,m-1} \rceil \quad | \quad A_{j,m} \in \mathbb{N}. \tag{A 5}$$

To estimate the amount of time that bus will have to stay at the bus stop $m$ for passenger boarding and alighting, a.k.a. *dwell time* $D_{j,m}$, we adopt the approach in [51] and the transit capacity and quality of service manual (TCQSM) [52]:

$$D_{j,m} = \theta_1 + \theta_2 \times B_{j,m} + \theta_3 \times A_{j,m}. \tag{A 6}$$

The parameter set $[\theta_1, \theta_2, \theta_3]$ represents the time spent for passenger boarding, alighting, and a fixed value for vehicle stopping and starting, respectively. Equation (A 6) is the formulation for a single-door bus system, where boarding and alighting occurs sequentially.

The departure time of bus $j$ from stop $m$ is calculated from the arrival time $t^a_{j,m}$ plus the time spent at stops for passenger boarding and alighting, or in other words the dwell time $D_m$:

$$t^d_{j,m} = t^a_{j,m} + D_{j,m}. \tag{A 7}$$

In BusSim, the bus $j$ is only allowed to leave the bus $m$ at time $t^d_{j,m}$, so this is also called the Leave_stop_time, as can be seen in figure 2.

If the status of bus $j$ is DWELLING, it is at a stop for passenger boarding and alighting. We then check if the next time step would be larger or equal to the leave stop time $t^d_{j,m}$. If it would, then the bus would start accelerating to leave the stop, otherwise it would stay for at least another time interval. Finally, if the status of the bus is FINISHED, then we would do nothing. The modelling process then moves to the next bus agent until the last bus, then the whole model moves to the next time step until the last time step.

BusSim-truth also assumes that parameters dynamically change over time by introducing an additional parameter $\xi$ to represent the change in passenger demand or surrounding traffic speed. For simplicity, we assume that a single, deterministic parameter $\xi$ can model these dynamic changes. In practice, it is possible, and more desirable, to use a time-dependent value of $\xi$ such that dynamic change is better captured, and multiple $\xi$ to model different changes. $\xi > 0$ represents an increase in passenger demand and traffic speed, and $\xi < 0$ represents otherwise. In this paper, the change in passenger demand or traffic speed is modelled as

$$V = V \cdot \left(1 - \frac{t}{T} \cdot \frac{100}{\xi}\right) \tag{A 8}$$

and

$$\text{Arr}_m = \text{Arr}_m \cdot \left(1 - \frac{t}{T} \cdot \frac{100}{\xi}\right). \tag{A 9}$$

A positive value of $\xi$ in equation (A 9) gradually reduces the surrounding traffic speed $V$ and increases the arrival rate $\text{Arr}_m$, which would lead to more bus delays and congestion.

# Appendix B. Cross entropy method for parameter calibration

This appendix describes the pseudocode for the cross entropy method for normal distribution [38].

---

**Algorithm 1.** Cross-entropy method for normal distribution.

---

1 Set $p = (\mu_1, \sigma_1, \mu_2, \sigma_2, ..., \mu_K, \sigma_K)$ %Initial distribution parameters

2 Set $M$ %Number of stops

3 Set $T$ %Maximum iteration number

4 Set $I$ %Maximum iteration number

5 Set $\rho$ %Set selection ratio

6 **for** $t$ from 1 to $T$ **do** $T$ **do**

7 %Main CEM loop

8 **for** $i$ from 1 to $I$ **do**

9 Draw $y^{(i)}$ from $N(\mu, \sigma)$ %Draw $I$ samples

10 Compute $f^i := f(y^{(i)}$

11 **end**

12 Sort $f^i$-values %Order by decreasing magnitude

13 $\gamma \leftarrow f_{\rho.I}$ %Set threshold

14 $L_\gamma \leftarrow \{y^{(i)} | f(y^{(i)}) \leq \gamma$ %Collect elite samples

15 $\mu'_j = \dfrac{1}{L_\gamma} \sum_{i=1}^{l_\gamma} \mu_{i,j}$ %Update $\mu$

16 $\sigma'_j = \dfrac{1}{L_\gamma} \sum_{i=1}^{l_\gamma} \sigma_{i,j}$ %Update $\sigma$

17 $\mu_j \leftarrow \alpha \mu'_j + (1 - \alpha)\mu_j$ %Update with step size $\alpha$

18 $\sigma_j \leftarrow \alpha \sigma'_j + (1 - \alpha)\sigma_j$ %Update with step size $\alpha$

19 **end**

---

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
