## [Reviewer comments · Royal Society Open Science]

Review History

RSOS-191074.R0 (Original submission)

Review form: Reviewer 1 (Jang Won Bae)

Is the manuscript scientifically sound in its present form?

Yes

Are the interpretations and conclusions justified by the results?

No

Is the language acceptable?

Yes

Do you have any ethical concerns with this paper?

No

Have you any concerns about statistical analyses in this paper?

No

Recommendation?

Major revision is needed (please make suggestions in comments)

Comments to the Author(s)

This paper introduces a combination of agent-based models and data assimilation. Specifically, the proposed framework consists of agent-based modeling and particle filter, and it is to increase the accuracy of model-based prediction.

After reading this paper, this reviewer comes up with a critical questions: Can assimilating with the simulation results (called real-time data) result in better prediction results? My answer is No. Because, as the authors said, a model is not the real-world, but its one facet. What is worse, the simulation model, used to generate the real-time data, seems not validated with the real-data. This question is very critical to this paper, so this reviewer expect that the authors should provide a proper answer to this question.

Other comments on the paper are listed as below:

1. Combination of agent-based model and data assimilation is not a new idea. The authors should see more references about this issue: Jang Won Bae, Euihyun Paik, Dong-oh Kang, Junyoung Jung, and Chun-Hee Lee. 2018. Simulation framework for self-evolving agent-based models: a case study of housing market model. In Proceedings of the 2018 Winter Simulation Conference (WSC '18). IEEE Press, Piscataway, NJ, USA, 1120-1131.
2. Because the data quality problem, the authors said that machine learning methods are not applicable to the problems the paper considers. However, the data with low quality also make difficult to be used for building simulation model. Do the authors have another reason? or provide more explanation?
3. Generally, a separation between static and dynamic models comes from changing model states over time, not model parameters. Therefore, for less confusion, the authors should define new terms for their own usage.
4. Model validation (i.e., comparing simulation results with the associated real-data) of BusSim-truth should be provided.
5. Data assimilation could occur when new real-data has been observed. In this paper, however, the real-time data provided from a simulation model is not in the case.
6. Why BusSim-deterministic and -stochastic are needed? when the parameters are calibrated, it seems enough to use BusSim-truth.
7. Why real-time data is used to data assimilation, not using historical one?
8. Figure 3 and Chapter 4(b) are better to be removed.
9. Traffic speed, a parameter to be calibrated, is not introduced in Table 1
10. It seems that the authors are confusing on model parameter and state; for example, the traffic speed seems to be a model state, not a parameter. There needs more clear cut between model parameter and state
11. on page 8, one equation may be removed (see line 3)
12. The first column of Table 1 seems to be missing.
13. No explanation about initially setting V_t
14. On page 12, The text says that $\max\text{Demand}$ equals 3, but not in the caption of Figure 4
15. Model verification is provided, but not model validation.
16. Sensitivity analysis seems not along to the context. It is better to be removed.

Review form: Reviewer 2

Is the manuscript scientifically sound in its present form?

Yes

Are the interpretations and conclusions justified by the results?

Yes

Is the language acceptable?

Yes

Do you have any ethical concerns with this paper?

No

Have you any concerns about statistical analyses in this paper?

No

Recommendation?

Major revision is needed (please make suggestions in comments)

Comments to the Author(s)

The paper attempts to address one of the most important issues in agent-based modeling, mainly how to update the model based on real-time data. In general, the authors describe a novel effort where they combine a Data Assimilation algorithm with agent-based modeling. This has been tried in the past to some degree, but the authors expand on that research.

The authors could also talk about ABM models that deal with transportation issues and/or bus routing. For example the company Anylogic (ABM software provider) has modelled such issues. Maybe the value of the paper could be expanded if the authors used (or even compared) the results of the BusSim-truth model.

There are several aspects that I do not completely agree with the authors in their methodology (this could also mean that more explanations are necessary). First, the application seems limited since the method depends only on the previous step. Why is that? What happens if this assumption is abandoned? Does the proposed hybridization of DA and ABM works? Maybe the authors should justify a little more their methodological assumptions.

Similarly, in the development of the BusSim-Truth model, I have some objections. I can understand the need to keep it simple enough to reduce computational complexity, but one important aspect of the bus routing/planning etc. is that not only buses are affected by the traffic volume, but they also affect it. For example, a bus when it is stopped could create abnormalities in the traffic flow in the surrounding area. The authors either do not consider it in the development of the model, or they omit to describe it. In any case, I believe that they should justify why they made that particular choice/assumption. In the best-case scenario, they could incorporate the relation in the model, and compare the results with the one they already describe.

More explanations of the results could greatly increase the value of the paper. For example why the observed differences? Does the choice of the DA algorithm matter? How the proposed DA technique compares to a simple Kahlman filter?

Finally, the paper is excellently written and well presented. I have observed only minor two mistakes:

Page 3/21: "but also for its tractability - there are many fewer interactions..."

Page 4/21 "...and DA in dealing the with stochastic and dynamic nature..."

Decision letter (RSOS-191074.R0)

16-Sep-2019

Dear Dr Kieu,

The editors assigned to your paper ("Dealing with uncertainty in agent-based models for short-term predictions") have now received comments from reviewers. We would like you to revise your paper in accordance with the referee and Associate Editor suggestions which can be found below (not including confidential reports to the Editor). Please note this decision does not guarantee eventual acceptance.

Please submit a copy of your revised paper before 09-Oct-2019. Please note that the revision deadline will expire at 00.00am on this date. If we do not hear from you within this time then it will be assumed that the paper has been withdrawn. In exceptional circumstances, extensions may be possible if agreed with the Editorial Office in advance. We do not allow multiple rounds of revision so we urge you to make every effort to fully address all of the comments at this stage. If deemed necessary by the Editors, your manuscript will be sent back to one or more of the original reviewers for assessment. If the original reviewers are not available, we may invite new reviewers.

- Data accessibility

If you wish to submit your supporting data or code to Dryad (<http://datadryad.org/>), or modify your current submission to dryad, please use the following link:
<http://datadryad.org/submit?journalID=RSOS&manu=RSOS-191074>

- **Competing interests**

- **Authors' contributions**

- **Acknowledgements**

- **Funding statement**

on behalf of Dr Danica Vukadinovic Greetham (Associate Editor) and Marta Kwiatkowska (Subject Editor)
openscience@royalsociety.org

Associate Editor's comments (Dr Danica Vukadinovic Greetham):

Based on the received reviews, we invite you to revise the paper according to the referees' suggestions.

Comments to Author:

Reviewers' Comments to Author:

Reviewer: 1

Comments to the Author(s)

This paper introduces a combination of agent-based models and data assimilation. Specifically, the proposed framework consists of agent-based modeling and particle filter, and it is to increase the accuracy of model-based prediction.

After reading this paper, this reviewer comes up with a critical questions: Can assimilating with the simulation results (called real-time data) result in better prediction results? My answer is No. Because, as the authors said, a model is not the real-world, but its one facet. What is worse, the simulation model, used to generate the real-time data, seems not validated with the real-data. This question is very critical to this paper, so this reviewer expect that the authors should provide a proper answer to this question.

Other comments on the paper are listed as below:

1. Combination of agent-based model and data assimilation is not a new idea. The authors should see more references about this issue: Jang Won Bae, Euihyun Paik, Dong-oh Kang, Junyoung Jung, and Chun-Hee Lee. 2018. Simulation framework for self-evolving agent-based models: a case study of housing market model. In Proceedings of the 2018 Winter Simulation Conference (WSC '18). IEEE Press, Piscataway, NJ, USA, 1120-1131.
2. Because the data quality problem, the authors said that machine learning methods are not applicable to the problems the paper considers. However, the data with low quality also make difficult to be used for building simulation model. Do the authors have another reason? or provide more explanation?
3. Generally, a separation between static and dynamic models comes from changing model states over time, not model parameters. Therefore, for less confusion, the authors should define new terms for their own usage.
4. Model validation (i.e., comparing simulation results with the associated real-data) of BusSim-truth should be provided.
5. Data assimilation could occur when new real-data has been observed. In this paper, however, the real-time data provided from a simulation model is not in the case.
6. Why BusSim-deterministic and -stochastic are needed? when the parameters are calibrated, it seems enough to use BusSim-truth.
7. Why real-time data is used to data assimilation, not using historical one?
8. Figure 3 and Chapter 4(b) are better to be removed.
9. Traffic speed, a parameter to be calibrated, is not introduced in Table 1
10. It seems that the authors are confusing on model parameter and state; for example, the traffic speed seems to be a model state, not a parameter. There needs more clear cut between model parameter and state
11. on page 8, one equation may be removed (see line 3)
12. The first column of Table 1 seems to be missing.
13. No explanation about initially setting V_t
14. On page 12, The text says that $\max\text{Demand}$ equals 3, but not in the caption of Figure 4
15. Model verification is provided, but not model validation.
16. Sensitivity analysis seems not along to the context. It is better to be removed.

Reviewer: 2

Comments to the Author(s)

The paper attempts to address one of the most important issues in agent-based modeling, mainly how to update the model based on real-time data. In general, the authors describe a novel effort where they combine a Data Assimilation algorithm with agent-based modeling. This has been tried in the past to some degree, but the authors expand on that research.

The authors could also talk about ABM models that deal with transportation issues and/or bus routing. For example the company Anylogic (ABM software provider) has modelled such issues. Maybe the value of the paper could be expanded if the authors used (or even compared) the results of the BusSim-truth model.

There are several aspects that I do not completely agree with the authors in their methodology (this could also mean that more explanations are necessary). First, the application seems limited since the method depends only on the previous step. Why is that? What happens if this assumption is abandoned? Does the proposed hybridization of DA and ABM works? Maybe the authors should justify a little more their methodological assumptions.

Similarly, in the development of the BusSim-Truth model, I have some objections. I can understand the need to keep it simple enough to reduce computational complexity, but one important aspect of the bus routing/planning etc. is that not only buses are affected by the traffic volume, but they also affect it. For example, a bus when it is stopped could create abnormalities in the traffic flow in the surrounding area. The authors either do not consider it in the development of the model, or they omit to describe it. In any case, I believe that they should justify why they made that particular choice/assumption. In the best-case scenario, they could incorporate the relation in the model, and compare the results with the one they already describe.

More explanations of the results could greatly increase the value of the paper. For example why the observed differences? Does the choice of the DA algorithm matter? How the proposed DA technique compares to a simple Kahlman filter?

Finally, the paper is excellently written and well presented. I have observed only minor two mistakes:

Page 3/21: "but also for its tractability – there are many fewer interactions..."

Page 4/21 "...and DA in dealing the with stochastic and dynamic nature...."

Author's Response to Decision Letter for (RSOS-191074.R0)

See Appendix A.

RSOS-191074.R1 (Revision)

Review form: Reviewer 1 (Jang Won Bae)

Is the manuscript scientifically sound in its present form?

Yes

Are the interpretations and conclusions justified by the results?

Yes

Is the language acceptable?

Yes

Do you have any ethical concerns with this paper?

No

Have you any concerns about statistical analyses in this paper?

No

Recommendation?

Accept with minor revision (please list in comments)

Comments to the Author(s)

This revision is well prepared, so this reviewer is almost satisfied with it. The following lists are the remained comments, and this reviewer hope to consider them for improving the paper quality:

1. This comment was not properly conveyed to the authors, so this reviewer asks again; The author said that the historical data is used for the parameter calibration, and the real-time data is used in the data assimilation. Then, the result of the data assimilation compared with the real-time data for the evaluation. It seems analogy that the training data is used in the model validation. Do the authors apply any kinds of cross validations?
2. In Section 2, this reviewer considers that machine learning and simulation are not compared with same weights; because listing the weakness of the machine learning and the benefits of the simulation, readers may misread the points. the balanced comparison should be backed up in the text. Also, the analytic model is not relevant to reveal the underlying process.
3. The authors argue that Figure 3 is required in this paper. If so, Figure 3 needs more explanations on the caption and the texts. Currently, This reviewer failed to understand what the y-axis means: distance is a relative metric, so it need to be explained.

Review form: Reviewer 3

Is the manuscript scientifically sound in its present form?

Yes

Are the interpretations and conclusions justified by the results?

Yes

Is the language acceptable?

Yes

Do you have any ethical concerns with this paper?

No

Have you any concerns about statistical analyses in this paper?

No

Recommendation?

Accept with minor revision (please list in comments)

Comments to the Author(s)

I have two minor comments with the paper:

- 1) The authors chose not to include noise in the BusSim-Truth model (or equivalently) to the generated data. I believe that this should be in the analysis, because let's say for example that something "extreme" occurs and the available "real" data do not make any sense (for example extreme malfunctions of the positioning services). What would happen in this case? What would be the behavior of the model? Does the algorithm account for such an event?
- 2) I have observed some minor mistakes. P3 line 21,
P3.line 57 "more complex phenom*a*"
p6.line 27 "it*s* paramaters"

Decision letter (RSOS-191074.R1)

15-Nov-2019

Dear Dr Kieu,

On behalf of the Editors, I am pleased to inform you that your Manuscript RSOS-191074.R1 entitled "Dealing with uncertainty in agent-based models for short-term predictions" has been accepted for publication in Royal Society Open Science subject to minor revision in accordance with the referee suggestions. Please find the referees' comments at the end of this email.

The reviewers and Subject Editor have recommended publication, but also suggest some minor revisions to your manuscript. Therefore, I invite you to respond to the comments and revise your manuscript.

- Ethics statement

- Data accessibility

If you wish to submit your supporting data or code to Dryad (<http://datadryad.org/>), or modify your current submission to dryad, please use the following link:
<http://datadryad.org/submit?journalID=RSOS&manu=RSOS-191074.R1>

- **Competing interests**

- **Authors' contributions**

- **Acknowledgements**

- **Funding statement**

Because the schedule for publication is very tight, it is a condition of publication that you submit the revised version of your manuscript before 24-Nov-2019. Please note that the revision deadline will expire at 00.00am on this date. If you do not think you will be able to meet this date please let me know immediately.

- 1) A text file of the manuscript (tex, txt, rtf, docx or doc), references, tables (including captions) and figure captions. Do not upload a PDF as your "Main Document".
- 2) A separate electronic file of each figure (EPS or print-quality PDF preferred (either format should be produced directly from original creation package), or original software format)
- 3) Included a 100 word media summary of your paper when requested at submission. Please ensure you have entered correct contact details (email, institution and telephone) in your user account
- 4) Included the raw data to support the claims made in your paper. You can either include your data as electronic supplementary material or upload to a repository and include the relevant DOI within your manuscript
- 5) Archived your GitHub code within Zenodo. Instructions on how to do this can be viewed here: <https://guides.github.com/activities/citable-code/>. The Zenodo DOI should be included within your manuscript and added to your data availability statement (for example: "Source code are available within GitHub [URL] and have been archived within Zenodo [doi:XXXXXX])
- 5) All supplementary materials accompanying an accepted article will be treated as in their final form. Note that the Royal Society will neither edit nor typeset supplementary material and it will be hosted as provided. Please ensure that the supplementary material includes the paper details where possible (authors, article title, journal name).

Kind regards,
Lianne Parkhouse
Editorial Coordinator
Royal Society Open Science
openscience@royalsociety.org

on behalf of Dr Danica Vukadinovic Greetham (Associate Editor) and Marta Kwiatkowska (Subject Editor)
openscience@royalsociety.org

Associate Editor Comments to Author (Dr Danica Vukadinovic Greetham):

Based on the two reviews and after reading the revision, I suggest to accept with minor revision.

Some further minor comments:

- p3 line 21 "in dealing the with" 'dealing with the'
- p3 line 40 "there are an array" 'there is'
- p9 line 28 "to dynamically optimised" , 'optimise'
- p9 line 51 "the best guest" , 'guess'
- p10 line30 should have full-stop after 'distribution'
- p20line 22: (author?) should be removed

Reviewer comments to Author:

Reviewer: 1

Comments to the Author(s)

This revision is well prepared, so this reviewer is almost satisfied with it. The following lists are the remained comments, and this reviewer hope to consider them for improving the paper quality:

1. This comment was not properly conveyed to the authors, so this reviewer asks again; The author said that the historical data is used for the parameter calibration, and the real-time data is used in the data assimilation. Then, the result of the data assimilation compared with the real-time data for the evaluation. It seems analogy that the training data is used in the model validation. Do the authors apply any kinds of cross validations?
2. In Section 2, this reviewer considers that machine learning and simulation are not compared with same weights; because listing the weakness of the machine learning and the benefits of the simulation, readers may misread the points. the balanced comparison should be backed up in the text. Also, the analytic model is not relevant to reveal the underlying process.
3. The authors argue that Figure 3 is required in this paper. If so, Figure 3 needs more explanations on the caption and the texts. Currently, This reviewer failed to understand what the y-axis means: distance is a relative metric, so it need to be explained.

Reviewer: 3

Comments to the Author(s)

I have two minor comments with the paper:

- 1) The authors chose not to include noise in the BusSim-Truth model (or equivalently) to the generated data. I believe that this should be in the analysis, because let's say for example that something "extreme" occurs and the available "real" data do not make any sense (for example extreme malfunctions of the positioning services). What would happen in this case? What would be the behavior of the model? Does the algorithm account for such an event?
- 2) I have observed some minor mistakes. P3 line 21,
P3.line 57 "more complex phenomen*a*"
p6.line 27 "it*s* paramaters"

Author's Response to Decision Letter for (RSOS-191074.R1)

See Appendix B.

Decision letter (RSOS-191074.R2)

28-Nov-2019

Dear Dr Kieu,

It is a pleasure to accept your manuscript entitled "Dealing with uncertainty in agent-based models for short-term predictions" in its current form for publication in Royal Society Open Science. The comments of the reviewer(s) who reviewed your manuscript are included at the foot of this letter.

on behalf of Dr Danica Vukadinovic Greetham (Associate Editor) and Marta Kwiatkowska (Subject Editor)
openscience@royalsociety.org

Appendix A

Response to reviewers

We would like to thank the two reviewers for their valuable comments. Our responses to each comment are below (with responses for Reviewer 2 from page 7).

Reviewer 1

Question 1.1. *Critical question: Can assimilating with the simulation results (called real-time data) result in better prediction results? My answer is No. Because, as the authors said, a model is not the real-world, but its one facet. What is worse, the simulation model, used to generate the real-time data, seems not validated with the real-data.*

As the reviewer suggests, this is a very important question. There are two issues here, that we will deal with in turn: (i) model structure and (ii) validation.

With regards to model structure, the reviewer rightly points out that the model is not the real world. We are well aware that, with apologies for the much over-used aphorism “all models are wrong”. Here, a model of the bus system is required that is sufficiently realistic to recreate common features of the bus system (such as ‘bus bunching’) but is simple enough to allow us to fully understand the dynamics. Section 2 (from page 4 specifically) explains that “BusSim-truth will ... replicate popular phenomenon in bus operations such as bus bunching (two buses of the same line arrive at the same bus stop at the same time)”. Building a more realistic model is beyond the scope of the paper (the aim is *not* to create an advanced, complicated model of bus operations) and would detract from the core aim – which is to test the value of data assimilation in improving predictions in an agent-based model – by making it harder to understand the underlying dynamics that drive the system. We agree with the reviewer that the model is not a ‘digital twin’ of the real world, but we argue that it is able to represent the key features of the real system in sufficient detail to allow us to test the value of a data assimilation algorithm.

With regards to model validation (ii), the reviewer is correct to raise the issue. We agree that with an unvalidated model it would be extremely difficult to argue that the results have any implications for the real-world. Section 4(b) (page 11) provides a ‘face’ validation of the model by demonstrating that it is able to replicate commonly observed features of a real bus system.

In response to the reviewers comment, we have added a new figure (Figure 4) to Section 4(b).

Question 1.2. *Combination of agent-based model and data assimilation is not a new idea. The authors should see more references about this issue: Jang Won Bae, Euihyun Paik, Dong-oh Kang, Junyoung Jung, and Chun-Hee Lee. 2018. Simulation framework for self-evolving agent-based models: a case study of housing market model. In Proceedings of the 2018 Winter Simulation Conference (WSC '18). IEEE Press, Piscataway, NJ, USA, 1120-1131.*

Thank you for highlighting this relevant paper, we make mention of it in Section 2 (page 4 specifically).

Question 1.3. *Because the data quality problem, the authors said that machine learning methods are not applicable to the problems the paper considers. However, the data with low quality also make difficult to be used for building simulation model. Do the authors have another reason? or provide more explanation?*

This is a valid point, but stems from a poor explanation in the paper. We did not intend to argue that simulation models will perform better than machine learning methods with poor quality data. Rather the point was that a simulation model can help to infer some of the unobserved features of the system that are not present in the available data. This is not something that machine learning methods are *typically* designed to do.

We have amended the first paragraph in Section 2 (page 3) to better explain this:

While machine learning methods are generally efficient in real time, they are solely reliant on the quality and quantity of the available data. Even with high-resolution datasets that

record accurate spatio-temporal bus locations, there are an array of additional features that are not recorded in the observed data (such as the downstream population waiting for a bus) so the full complexity of the system will never be captured.

Instead, analytical and simulation models of bus routes have been proposed that aim to reproduce the *underlying processes* in bus operations, rather than attempting to identify direct mappings between inputs and outputs. One of the earliest successes in simulating a simple bus systems was from Cellular Automata modelling [10, 22, 32, 39].

Question 1.4. *Generally, a separation between static and dynamic models comes from changing model states over time, not model parameters. Therefore, for less confusion, the authors should define new terms for their own usage.*

We have revised the relevant paragraph in the Section 2 (Research Problem and Related works) to address your comment.

One way for these models to fit better to the observed data is to adjust the model parameters until the model satisfies some predetermined criteria. This parameter adjustment process is often referred to as *parameter calibration*. Popular optimisation techniques include simulated annealing [41], genetic algorithms, [18, 34], and approximate Bayesian computation [15]. Parameter calibration, especially with ABMs, is often only implemented once, and therefore cannot account for any changes that may take place within the system during run time. In fact, the real bus operation is certainly *dynamic*, where the system states are changing over time, e.g. changing traffic condition or passenger demand. In real time, there are also other uncertainty about the bus operations that comes from the lack of information regarding the current system states. Examples of such unobservable system states include the number of passengers who are waiting at downstream stops or the number who plan to get off the bus, and the surrounding traffic conditions. The lack of information about these factors means that any model of bus operation in real time will have to make assumptions thereby introducing errors in their predictions.

Question 1.5. *Model validation (i.e., comparing simulation results with the associated real-data) of BusSim-truth should be provided.*

We agree with the need for validation and address this in our response to question 1.1.

Question 1.6. *Data assimilation could occur when new real-data has been observed. In this paper, however, the real-time data provided from a simulation model is not in the case.*

The assertion made by the review is correct, we do not yet attempt to apply these algorithms to real-world data. We do, however, apply them to pseudo-realistic real-time data, as a precursor to an application to a real system. This pseudo-truth experimental framework has been used regularly in this context.

Ultimately the aim of this work is to apply these methods to real systems, but as a first step, we need to test that the methods are appropriate for an abstract system that we *fully understand*. That is the motivation of creating a model to generate pseudo-truth data, rather than using real data directly. These experiments are an essential part of testing how well the model performs in an environment in which we can fully quantify the error. This accurate quantification is not possible in a real-world scenario, because the *true* state of the system can never be known precisely.

These points are made explicitly in the introduction (e.g. on page 1) but we have added some additional text to the conclusion to clarify the position:

Ultimately the methods will be applied to real data from real systems, but currently hypothetical 'pseudo-truth' data are generated to test the algorithms in an experimental environment as per the 'identical twin' approach.

Question 1.7. *Why BusSim-deterministic and -stochastic are needed? when the parameters are calibrated, it seems enough to use BusSim-truth.*

Thank you for raising this point. We have rewritten the 2 (Research Problem and Related works) to address your comment. A mistake in Figure 1 has also been fixed.

In practice, one would develop simulation models of bus systems that aim to replicate the real bus operations by providing outputs that are as close as possible to some historical bus data. This paper, instead, follows a 'pseudo-truth' experiment framework, similar to [51]. The experiment data to be used will be 'synthetic', or generated from simulation instead of using real data. The reason is that real data often comes with noise that hides the true state of the bus route (e.g. noise from GPS data). A simulated synthetic data would enable us to control the level of noise in the data, and to evaluate the modelling results against the ground truth rather than noisy data. Figure 1 shows the workflow of this study.

1. Generate pseudo-truth data and model calibration

2. Real-time data assimilation

Any simulation model, in practice, is essentially an imperfect replication of reality. For instance, the real bus operation is both *dynamic* (the system states are changing over time) and *stochastic* (there are inherent randomness in the system). There are many bus simulation models in literature that are *static* (the system states remain the same over time) and *deterministic* (the same parameters return exactly the same simulation outputs). Taking this into consideration, we develop a framework (illustrated in Figure 1) where the model that will be used to generate the synthetic 'pseudo-truth' data is both *dynamic* and *stochastic*, to be as close as possible to the reality. We will refer to this model as *BusSim-truth*. The developed framework will then be evaluated on two simpler variations

of this model, knowing that they would not be able to perfectly represent the dynamics in BusSim-truth. This is similar to the practice where no simulation model of bus operations would be able to completely replicate the dynamics of real bus systems. Both simpler models are static (i.e. they both have states that are unchanged over time) but one of them is *deterministic* and another one is *stochastic*:

- **BusSim-deterministic.** This model evolves exactly the same way in each model run;
- **BusSim-stochastic.** This model is stochastic, e.g. the numbers of people waiting at bus stops is drawn from a random distribution

The study workflow generally consists of 2 major steps. It starts with the development of the *BusSim-truth* model. Two sets of pseudo-truth data will be generated. The first represents 'historical' GPS data, which are essentially the outputs of multiple runs of the same BusSim-truth model with the same predefined set of parameters. The GPS data will be slightly different each time the model is run because BusSim-truth is stochastic and dynamic. The second set of data represent a single run of BusSim-truth, also using the same set of parameters. These data will represent synthetic 'real-time' GPS data and will be used to conduct data assimilation. This situation is similar to the reality, where transport companies collect data across multiple days to build up a 'history' of the behaviour of the bus system and subsequently use these data to calibrate models. The 'real-time' data represent the *current* state of the world. The BusSim-truth model is indeed not a perfect replication of reality, but is reasonably realistic, has the *dynamic* and *stochastic* features similar to a real bus system, and replicates popular phenomenon in bus operations such as bus bunching (where two buses of the same line arrive at the same bus stop at the same time).

As would be necessary in reality, BusSim-deterministic and BusSim-stochastic will first be calibrated against the synthetic 'historical' GPS data. In the second step of the study workflow, DA will be used in an attempt to update the states of the models to the 'real-time' GPS observations in order to produce more accurate short-term forecasts of the system behaviour.

Question 1.8. *Why real-time data is used to data assimilation, not using historical one?*

Two pseudo-real data sets are used and we do this to reflect the situation that would occur were this research applied to a real bus route. Both data sets are generated by the BusSim-Truth agent-based model. The first is a large volume of 'historical' data, which is analogous to the data sets that transport companies collect on a daily basis. Such data are used to calibrate the model parameters. The second set is used to simulate the pseudo-real-time operations of the bus network, i.e what would be happening in *real time*. The historical data could be used to do this as well, but that would weaken the experiments because the model would be trying to assimilate data that it had already been calibrated on. It is more rigorous to use a new set of data that the model has not seen before.

Figure 1 (page 5) outlines the workflow in detail, but we have also added more text to the paragraph that describes the workflow on page 19:

This situation is similar to the reality, where transport companies collect data across multiple days to build up a 'history' of the behaviour of the bus system and subsequently use these data to calibrate models. The 'real-time' data represents the *current* state of the world.

Question 1.9. *Figure 3 and Chapter 4(b) are better to be removed.*

We argue that there is value in Figure 3 because it makes it clear the difference between the ‘historical’ and ‘real time’ data. It also demonstrates the degree to which BusSim-Truth is stochastic, which is important because were it more or less stochastic then the data it generates would not be so similar to those of a real bus system.

Similarly, Section 4(b) is important because it demonstrates that the model behaves as expected under variations in passenger demand. In fact, we added Figure 4 and more explanation to Section 4(b) to address your earlier comments regarding validation of the model.

As the second reviewer has not also recommended removing these parts of the paper we feel that these sections offer a useful contribution to the research presented.

Question 1.10. *Traffic speed, a parameter to be calibrated, is not introduced in Table 1.*

Thank you for pointing this out. We have added the traffic speed V_t to the Table 1.

Question 1.11. *It seems that the authors are confusing on model parameter and state; for example, the traffic speed seems to be a model state, not a parameter. There needs more clear cut between model parameter and state.*

There is an ongoing debate between parameters, variables and model states. To clarify from our perspective, variables or model states are entities that are changing inherently within the model as it is running, e.g. the vector $O_t = [c_j^t \ s_j^t \ v_j^t \ Occ_j^t]$ in Equation A 10. We define parameters as ‘forcing inputs’ (see Conti et al. 2009 where the term is used for dynamic simulation models) that are used to influence model states. An example of these parameters in the vector $S_t = [Arr_m^t \ Dep_m^t \ V^t]$ in Equation 3.1. In the Parameter Optimisation step (Section 3b), we only calibrate the parameters in S_t (including the traffic speed parameter that you mentioned). In the Data Assimilation step (Section 3c), there parameters in S_t are also dynamically calibrated along with the model states in O_t .

We have revised the Section 3c to address your comment.

We can formulate an ABM as a state-space model $\dot{X}_t = f(X_t) + \epsilon_t$ and use data assimilation (DA) to dynamically optimise the model states and parameters with up-to-date data to reduce uncertainty. The state-space model is represented by a state-space vector X_t at time t , which contains all information of the current state of each agent in the model, and the important parameters that we want to dynamically optimised as the model is running.

$$X_t = [O_t \ S_t] \\ = \left[\begin{array}{ccccccc} c_j^t & s_j^t & v_j^t & Occ_j^t & Arr_m^t & Dep_m^t & V^t \end{array} \right] \quad (\text{A } 10)$$

Reference: S. Conti, J. P. Gosling, J. E. Oakley, A. O’Hagan, Gaussian process emulation of dynamic computer codes, *Biometrika*, Volume 96, Issue 3, September 2009, Pages 663–676, <https://doi.org/10.1093/biomet/asp028>

Question 1.12. *On page 8, one equation may be removed (see line 3)*

Page 8 has only one equation (Equation 3.2), which is not on the line 3. The authors think that this equation is useful for readers without optimisation background to understand the parameter optimisation problem.

Question 1.13. *The first column of Table 1 seems to be missing.*

Thank you for pointing this out. We have revised the Table 1 for better readability and to add the missing parameters/variables.

Question 1.14. *No explanation about initially setting V_t*

Thank you for pointing this out. We have added the information in Section 4a

Finally, the initial traffic speed is set at 14 m/s.

Question 1.15. *On page 12, The text says that $maxDemand$ equals 3, but not in the caption of Figure 4*

Thank you for your comment. We have fixed the mistake you mentioned.

The solid lines show the trajectory of buses at high and stochastic demand ($maxDemand$ equals 2), whereas the dashed lines are for low and deterministic demand ($maxDemand$ equals $minDemand$ and equals 0.5).

Question 1.16. *Model verification is provided, but not model validation.*

Agreed; we discuss model validation in our response to question 1.1.

Question 1.17. *Sensitivity analysis seems not along to the context. It is better to be removed.*

We're not entirely sure what the precise suggestion relates to here, but don't think that it would be appropriate to remove the sensitivity analysis. It forms an important part of the research process.

Reviewer 2

Question 2.1. *The authors could also talk about ABM models that deal with transportation issues and/or bus routing. For example the company Anylogic (ABM software provider) has modelled such issues. Maybe the value of the paper could be expanded if the authors used (or even compared) the results of the BusSim-truth model.*

This is a very good point; although they are not yet abundant there are some important examples of agent-based bus simulation that we have not mentioned. We have added some new text to Section 2 (page 3 specifically) that discusses relevant models and outlines how they compare to the *BusSim-Truth* model. We have stopped short of comparing the results directly though as this is beyond the scope of the paper. Although we argue that *BusSim-Truth* produces reasonable bus system behaviour (as discussed in the revised section on validation – Section 4(b)) the aim of the paper is not to build a leading bus simulation, rather a simple simulation is required that is able to capture some of the main features of the system. Future work will test these methods on more advanced systems and simulations.

To address your very constructive comment, we have revised the Section 2 by adding a literature review on ABMs of buses (page 3), and by rewriting the section about the focus of the paper. We have also added a short discussion about this to Section 5 (which has also now been renamed ‘Implications and Limitations’ to better reflect the attention to limitations as well as future opportunities).

Question 2.2. *There are several aspects that I do not completely agree with the authors in their methodology (this could also mean that more explanations are necessary). First, the application seems limited since the method depends only on the previous step. Why is that? What happens if this assumption is abandoned? Does the proposed hybridization of DA and ABM works? Maybe the authors should justify a little more their methodological assumptions.*

Thank you for your insightful comment. You’ve hit on one some of the difficulties with integrating ABM and DA. There are two issues with respect to the this question that we deal with in turn.

- (i) Firstly, the data assimilation step does not need to take place at every model iteration. The underlying models could iterate forward for any amount of time in between conducting data assimilation. Here we choose to apply the DA after every time step because it is reasonable to assume that the bus GPS data arrive in near real time (or at least in a time that is comparable to the size of a model simulation step).
- (ii) A more complicated difficulty is that, in effect, the underlying model needs to be Markovian. I.e. it needs to be able to take a prior state as input and advance the model forward to create a future state without any other information. In this application it is not difficult to make the model Markovian, but for more complicated agent-based models, especially those that incorporate complex (human?) decision making with agent states that depend on their history, making them Markovian may not be trivial.

We discuss the need for Markovian models in Section 2 (page 2 specifically) already, but haven’t discussed the fact that data assimilation occurs at every time step. To clarify this we have added some new text to Section 3(c) (page 10) and have also made some small adjustments to Figure 1 to make the time evolution clearer.

Question 2.3. *Similarly, in the development of the BusSim-Truth model, I have some objections. I can understand the need to keep it simple enough to reduce computational complexity, but one important aspect of the bus routing/planning etc. is that not only buses are affected by the traffic volume, but they also affect it. For example, a bus when it is stopped could create abnormalities in the traffic flow in the surrounding area. The authors either do not consider it in the development of the model, or they omit to describe it. In any case, I believe that they should justify why they made that particular choice/assumption. In the best-case*

scenario, they could incorporate the relation in the model, and compare the results with the one they already describe.

This is a valid point; in many bus systems the buses will affect the surrounding traffic. As the reviewer points out, the treatment of traffic in the model is relatively simple. In Section 3(a) we explain that:

Currently the traffic volume on the whole network is represented as a single dynamic parameter, although in practice it would be relatively simple to make the traffic volume heterogeneous across the network.

Therefore a more nuanced treatment of traffic in the model would first require the traffic to vary heterogeneously on each road link. This is not technically difficult to implement in the model, but would substantially increase the size and complexity of the model state space – rather than one parameter for the traffic (V^t in Equation 3.1) there would be one parameter per link. This increased state space size would increase the numbers of particles required significantly (probably exponentially) and although this is not technically difficult (it would still be within the bounds of computation on a high-performance computer) it would require a number of new experiments to be conducted. Only after this would it make sense to experiment with the impacts of buses influencing traffic.

In summary, although including the impacts of buses on traffic is an interesting and important further addition, we argue that it is beyond the scope of the paper. Nevertheless, we agree with the reviewer that it needs discussing at least, so we have added part of our argument above to the ‘limitations’ part of the re-written Section 5.

Question 2.4. *More explanations of the results could greatly increase the value of the paper. For example why the observed differences? Does the choice of the DA algorithm matter? How the proposed DA technique compares to a simple Kahlman filter?*

Thank you for this constructive comment. We have added some more analysis and explanations of the modelling results to the Section 4(c) to 4(e).

We have also explained the rationale of choosing Particle Filter versus other DA algorithms in the section 3(c), but a new paragraph has also been added to the Section 4(e) to address your comment:

As discussed, data assimilation (DA) is the chosen approach to enable the static BusSim-stochastic and BusSim-deterministic models to deal with the uncertainty from a system that is changing over time. Particle Filter (PF) is the specific DA algorithm being adopted in this paper, thanks to its ability to deal with non-linear, non-Gaussian models without analytical structure. The well-known high computation cost concern of the PF [34, 51] is not really an issue in this study because of the limited number of agents (only bus and bus stop agents).

Question 2.5. *Finally, the paper is excellently written and well presented. I have observed only minor two mistakes:*

- Page 3/21: “but also for its tractability – there are many fewer interactions...”
- Page 4/21: “... and DA in dealing the with stochastic and dynamic nature...”

Thank you for your feedback. We have also gone through the paper to fix other grammar/spelling mistakes.

Appendix B

Response to reviewers

We would like to thank the two anonymous reviewers, Dr Danica Vukadinovic Greetham (Associate Editor) and Marta Kwiatkowska (Subject Editor) for their valuable comments. We are also delighted that the current manuscript only needs minor revisions to be published at the Royal Society Open Science. Our responses to each comment are below (with responses for Reviewer 2 from page 2).

Associate Editor Comments

Thank you for accepting the paper. we have addressed all of your concerns in the current manuscript

Reviewer 1

Question 1.1. *This comment was not properly conveyed to the authors, so this reviewer asks again; The author said that the historical data is used for the parameter calibration, and the real-time data is used in the data assimilation. Then, the result of the data assimilation compared with the real-time data for the evaluation. It seems analogy that the training data is used in the model validation. Do the authors apply any kinds of cross validations?*

Thank you for your very constructive comments. We have been able to improve the clarity and quality of the paper thanks to your feedback. To address this comment, we have added some more explanation of the Figure 1 in Section 2 (p.5)

Any simulation model, in practice, is essentially an imperfect replication of reality. For instance, the real bus operation is both *dynamic* (the system states are changing over time) and *stochastic* (there is inherent randomness in the system). Recall that the objective of this paper is to improve the accuracy of short-term forecasts using Agent-Based Models by performing dynamic state estimation of the current system state. This is essentially the final product at time $t + 1$ in Figure 1. Improvements of the forecasts at time $t + 1$ is archived by improving the estimation of system state at time t , by using Data Assimilation to transform a 'Prior state vector' (pure models' estimates) to a 'Posterior state vector' (models' estimates combined with real-time data).

In short, the 'cross-validation' is done by comparing the 'Improved forecasts' at time $t + 1$ with the synthetic real-time data at time $t + 1$, with the results illustrated in Figure 7, 8, and 9. The real-time data therefore is always unseen to the models. For a video of the model in action, see: <https://github.com/leminhkieu/leminhkieu.github.io/blob/master/videos/bussim-pf-video.mp4>

Question 1.2. *In Section 2, this reviewer considers that machine learning and simulation are not compared with same weights; because listing the weakness of the machine learning and the benefits of the simulation, readers may misread the points. the balanced comparison should be backed up in the text. Also, the analytic model is not relevant to reveal the underlying process.*

Thank you for your comment. The fact that machine learning models will provide accurate and efficient predictions has now been discussed in Section 2 of the manuscript.

Question 1.3. *The authors argue that Figure 3 is required in this paper. If so, Figure 3 needs more explanations on the caption and the texts. Currently, This reviewer failed to understand what the y-axis means: distance is a relative metric, so it need to be explained.*

We have added more explanation of the Figure 3 to address your comment.

As described in Section 2, we use BusSim-truth to generate two sets of synthetic data: (1) ‘historical’ GPS data that simulate normal bus route operation over a number of days and are used for calibration; and (2) ‘real-time’ GPS data that represents a single run of the model and are used to represent the bus system *today*. These are visualised in the time-space diagram in Figure 3. Each coloured line shows the trajectory of one bus in the ‘historical’ GPS data. The bold black lines are another instance of the bus trajectory that we consider as the ‘real-time’ GPS data.

The x-axis shows the time of simulation from 0 to 6000 seconds, where multiple bus services can be found. The y-axis shows the distance from the first bus stop (distance equals zero) to the last stop (distance equals 40,000 m). Assuming that all buses start their service on-time, Figure 3 shows bunches of bus service (each with a different colour) and their associated synthetic real-time data (bold lines). As the BusSim-truth model is stochastic, there are a spread of trajectories within each service. This is similar to the reality where buses operate slightly differently on multiple days.

Reviewer 2

Question 2.1. *The authors chose not to include noise in the BusSim-Truth model (or equivalently) to the generated data. I believe that this should be in the analysis, because let's say for example that something "extreme" occurs and the available "real" data do not make any sense (for example extreme malfunctions of the positioning services). What would happen in this case? What would be the behavior of the model? Does the algorithm account for such an event?*

This is a very good point; in the current paper we decided not to add noise in the data, because GPS data nowadays has very good positioning accuracy that 5-10m noise in position accuracy did not show significant difference in model performance, when our bus route is 40,000m. We instead focus on the problem of uncertainty in this paper, when information is missing in real-time (e.g. traffic condition, number of passengers waiting at the downstream stops). However, we admit that the uncertainty from cases when the data are completely non-sensical has not been addressed in this paper. Data assimilation methods, similar to many other data-driven methods such as machine learning, require and rely on the quality of data. We assume that the data have a good quality and added this limitation of the current work to Section 5.

Data assimilation methods, similar to many other data-driven methods, require and rely on the quality of data. Another limitation of this work is with regard to the cases where the data have major discrepancy, that have not been addressed in this current framework. The current models, in principal, should still produce particles that are more reliable than data that are highly inaccurate. Future studies may look at developing a data quality check procedure into the framework in Figure 1 to allow skipping the data assimilation step if the data quality deteriorates to a certain value.

Question 2.2. *I have observed some minor mistakes. P3 line 21, P3.line 57 "more complex phenomen*a*" p6.line 27 "it*s* paramaters" .*

Thank you for your feedback. Various minor grammar and spelling mistakes have been addressed in the revised version of this paper.